# Differentially Private Graph Diffusion with Applications in Personalized PageRanks

**Rongzhe Wei**
Georgia Institute of Technology
`rongzhe.wei@gatech.edu`

**Eli Chien**
Georgia Institute of Technology
`ichien6@gatech.edu`

**Pan Li**
Georgia Institute of Technology
`panli@gatech.edu`

## Abstract

Graph diffusion, which iteratively propagates real-valued substances among the graph, is used in numerous graph/network-involved applications. However, releasing diffusion vectors may reveal sensitive linking information in the data such as transaction information in financial network data. Protecting the privacy of graph data is challenging due to its interconnected nature. This work proposes a novel graph diffusion framework with edge-level differential privacy guarantees by using *noisy diffusion iterates*. The algorithm injects Laplace noise per diffusion iteration and adopts a degree-based thresholding function to mitigate the high sensitivity induced by low-degree nodes. Our privacy loss analysis is based on Privacy Amplification by Iteration (PABI), which to our best knowledge, is the first effort that analyzes PABI with Laplace noise and provides relevant applications. We also introduce a novel $\infty$-Wasserstein distance tracking method, which tightens the analysis of privacy leakage and makes PABI practically applicable. We evaluate this framework by applying it to Personalized Pagerank computation for ranking tasks. Experiments on real-world network data demonstrate the superiority of our method under stringent privacy conditions.

## 1 Introduction

Graph diffusion, characterized by propagating signals across networks, is used in a variety of real-world applications. Variants of graph diffusion such as PageRank [1] and heat kernel diffusion [2] has revolutionized the domains such as web searching [3], community detection [4–7], network analysis [8, 9] and advancements in graph neural networks [10–13]. Despite their widespread applications, directly releasing diffusion vectors can inadvertently leak sensitive graph information and raise privacy concerns. Hoskins et al. [14] demonstrate that the access to a small subset of random walk-based similarities (e.g., commute times, personalized PageRank scores) could disclose significant portions of network's edges, a phenomenon known as *link disclosure* [15]. Such attacks, for instance, may enable advertisers to deploy invasive advertising tactics [16] or reveal sensitive transaction information within financial networks [17]. Consequently, it becomes critically important to design graph diffusion algorithms with privacy safeguards.

Differential privacy (DP) is recognized as a gold standard used for characterizing the privacy risk of data processing algorithms [18]. However, the inherently interconnected nature of graph-structured data renders the adaptation of DP to graphs non-trivial [19]. Previous studies often conduct the analysis of output sensitivity and adopt output perturbation to keep graph data private, which include the study on differentially private personalized PageRanks (PPRs) [20], and other relevant graph

algorithms such as max flow-min cut [21], graph sparsification [22], spectral analysis [23, 24]. However, output perturbation-based approaches often provide a less-than-ideal utility-privacy tradeoff. Numerous studies suggest that incorporating noise during the process, rather than at the output, can potentially enhance utility-privacy tradeoffs [25].

In this work, we introduce a graph diffusion framework that ensures edge-level DP guarantees based on noisy diffusion iterates. Our framework is the *first* to incorporate privacy amplification by iterations (PABI) technique [26] into graph diffusions. As graph diffusion can be viewed as iterating contraction maps in $\ell_1$ space, we adopt per-iterate Laplace noise due to its better performance than the Gaussian mechanism commonly adopted in previous PABI frameworks and provide new analysis dedicated to Laplace noise. We also propose a novel $\infty$-Wasserstein distance tracking analysis that can tighten the state-of-the-art PABI bound [27] that relies on the space diameter, which makes the bound valid for practical usage. Noticing diffusion from low-degree nodes may introduce high sensitivity, our framework also also proposes a theory-informed *degree-based thresholding function* at each step diffusion to improve the utility-privacy tradeoff. Lastly, we specialize our framework in the computation of PPR for node ranking tasks. Extensive experiments reveal the advantages of our framework over baselines, especially under stringent privacy requirements.

## 1.1 More Related Works

Extensive research has been dedicated to privacy protection within graphs, specifically in the release of graph statistics and structures under DP guarantees [28, 29]. The primary techniques to safeguard graph structures involve the Laplace and exponential mechanisms [30]. Early contributions by Nissim et al. [31] calibrated noise based on the smooth sensitivity of graph queries, expanding beyond output perturbation. Karwa et al. [32] improved the efficiency of privacy-preserving subgraph-counting queries by calibrating Laplace noise according to smooth sensitivity. Different from these methods, Zhang et al. [33] employed the exponential mechanism [34] to enhance privacy protections. Concurrently, Hay et al. [35] developed a constraint-based inference algorithm as a post-processing step to improve the quality of degree sequences derived from output perturbation mechanisms. Further, Kasiviswanathan et al. [36] used a top-down degree projection technique to limit the maximum degree in graphs, thus controlling the sensitivity of degree sequence queries. Additional efforts in privately releasing graph statistics include outlinks [37], cluster coefficients [38], graph eigenvectors [39], and edge weights [40].

In the realm of graph diffusion, the most related work to ours is by Epasto et al. [20], which focuses on releasing the PPR vector using *forward push* [41] and Laplace output perturbation. Some studies have shown that injecting noise during the process may offer better privacy-utility trade-offs compared to output perturbation methods [25] and our study compared with [20] provides another use case. Traditionally, the composition theorem [42] is used to track privacy guarantees for iterative algorithms, but it results in a bound that may diverge with the number of iterations. Recently, the technique of PABI [26, 27] was introduced to strengthen the privacy analysis of adding noise during the process, which demonstrates a non-divergent privacy bound for releasing final results if the iterations adopt contraction maps [27]. This substantially tightens the divergent bound given by the naive application of the DP composition theorem [43, 44]. Our framework also benefits from this advantage and we further adapt Altschuler et al.'s analysis [27] to incorporate the Laplace mechanism and provide a tightened space diameter tracking, which makes the bound practically applicable in the graph diffusion applications. Besides, works that share a similar spirit in leveraging PABI have been conducted for other scenarios including machine unlearning [45] and improving hidden state DP [46].

## 2 Preliminaries

Let $\mathcal{G} = (\mathcal{V}, \mathcal{E})$ represent an undirected graph, where $\mathcal{V}$ is the set of nodes and $\mathcal{E}$ is the set of edges, equipped with an adjacency matrix $\mathbf{A} \in \{0, 1\}^{n \times n}$, where $n$ denotes the total number of nodes, i.e., $n = |\mathcal{V}|$. By establishing an order for the nodes within the graph, we denote $\mathbf{d} = [d_1, d_2, ..., d_n]^T$ as the degree vector. Additionally, let $\mathbf{D} = \text{diag}(\mathbf{d})$ and $\mathbf{e}_i$ signifies the $i$-th standard basis. We denote $\mathcal{L}(\mathbf{0}, \sigma)$ and $\mathcal{N}(\mathbf{0}, \sigma^2 \mathbf{I})$ as the zero mean Laplace and Gaussian distributions, respectively. We define the set $[n] = \{1, 2, \ldots, n\}$, and $X_{i:j}, i, j \in \mathbb{Z}_+, i \le j$ as joint couple of $(X_i, X_{i+1}, \ldots, X_j)$.

**Graph Diffusion.** First, we introduce the concept of **Graph Diffusion** $\mathscr{D}$, which is commonly characterized by a series of diffusion map $\phi_k$ defined by the random walk matrix $\mathbf{P} = \mathbf{A}\mathbf{D}^{-1}$ [6, 12,

47]. Formally, We define the graph diffusion $\mathscr{D}(\mathbf{s})$ with the initial seed $\mathbf{s}$ as

$$\mathscr{D}(\mathbf{s}) = \lim_{K \to \infty} \mathbf{s}_K = \lim_{K \to \infty} \phi_K \circ \cdots \circ \phi_1(\mathbf{s}), \text{ where } \phi_k(\mathbf{x}) = (\gamma_{1,k}\mathbf{P} + \gamma_{2,k})\mathbf{x} + \gamma_{3,k}\mathbf{s}. \quad (1)$$

where $\mathbf{s} \in \mathbb{R}^{|\mathcal{V}|}$ is a stochastic vector on the graph, and $\gamma_{1,k} + \gamma_{2,k} + \gamma_{3,k} = 1$. Let $\gamma_{\max} = \max_k |\gamma_{1,k}| + |\gamma_{2,k}|$ denote the Lipschitz constant of the graph diffusion mapping, and $\gamma_{\max}^{(1)} = \max_k |\gamma_{1,k}|$ denote the maximum diffusion coefficient. $\mathbf{s}_K$ is the diffusion vector at time $K$. The essence of a diffusion process is to model how an initial vector $\mathbf{s}$ propagates through the graph over time. Coefficient $\gamma_{i,k}$'s control how different resources contribute to the $k$th step diffusion. When taking $\gamma_{1,k} = 1 - \gamma_{3,k} = \beta$ and $\gamma_{2,k} = 0$, Eq. (1) is recognized as the PageRank Diffusion [48] with teleport probability $1 - \beta$. The Exponential kernel diffusion, which includes the specific case of the Heat Kernel diffusion [2], can also be characterized with the composition of diffusion mappings via the infinitely divisible property [49].

**Personalization.** Personalized graph diffusions, tailored to individual nodes or localized neighborhoods, play a crucial role in many real-world applications. These include recommendation systems [50], where personalized diffusions improve suggestion relevance, community detection for identifying subgroups within larger networks [7], targeted marketing strategies for enhancing campaign effectiveness [51]. These diffusions are defined by setting the graph diffusion vector $\mathbf{s}$ as $\mathbf{e}_i$ for an individual node or $\mathbf{s} = \sum_{i \in S} \mathbf{e}_i / |S|$ for a neighborhood set $S$. In this paper, we primarily discuss the single-node case while our analysis can be generalized to a set of seed nodes.

**Privacy Definition.** Differential Privacy (DP) [18,42] is widely recognized as the standard framework for providing formal privacy guarantees for algorithms that process sensitive data. This framework has further been extended under Rényi divergence [44]. Its principles have been applied to safeguard sensitive structures within graph algorithms, an metric noted as Edge-level Rényi Differential Privacy (RDP). Details on the conversion from RDP to DP are elaborated in App. E.

**Definition 1** (Edge-level RDP [52,53]). *A randomized graph algorithm $\mathcal{A}$ is $(\alpha, \epsilon)$-edge-level RDP if for any adjacent graphs $\mathcal{G}, \mathcal{G}'$ that differs in a single edge, we have $\mathcal{D}_\alpha(\mathcal{A}(\mathcal{G})\|\mathcal{A}(\mathcal{G}')) \le \epsilon$, where the Rényi Divergence $\mathcal{D}_\alpha(X\|Y) = \frac{1}{\alpha-1}\log \mathbb{E}_{x\sim\nu}\left(\frac{\mu(x)}{\nu(x)}\right)^\alpha$ with $X \sim \mu, Y \sim \nu$.*

More practical cases find that the seed node (user) of personalized graph diffusion algorithms is already aware of their direct connections within the network, such as one's friend list in social networks, and one's transaction record in financial networks, and therefore protecting the edges directly attached to the seed node becomes unnecessary. Instead, the focus of privacy protection shifts towards obscuring the connections between the remaining nodes. To address this specific need, we follow the previous study [20] and introduce Personalized Edge-level RDP.

**Definition 2** (Personalized Edge-level RDP [20,54]). *Consider a graph $\mathcal{G}$ and a personalized graph algorithm $\mathcal{A}$. The algorithm $\mathcal{A}$ satisfies personalized $(\alpha, \epsilon)$-edge-level RDP if for any node $v$ as the seed node in $\mathcal{G}$, and for any graph $\mathcal{G}'$ adjacent to $\mathcal{G}$ differing by one edge not incident to $v$, we have $\mathcal{D}_\alpha(\mathcal{A}(\mathcal{G}, v)\|\mathcal{A}(\mathcal{G}', v)) \le \epsilon$.*

## 3 Methodology

This study centers on a category of $\gamma_{\max} < 1$ Lipschitz continuous graph diffusions, encompassing prevalent techniques such as PageRank [1] and PPR [55]. It is noted that each diffusion map $\phi_k$ within graph diffusion maintains the total mass of the diffusion vector $\mathbf{s}_k$, owing to the condition $\sum_{i \in [3]} \gamma_{i,k} = 1$ and the property of the random walk matrix $\mathbf{P}$ being a left stochastic matrix. This observation entails that the diffusion map $\phi_k$ in Eq. (1) constitutes a strictly contraction map in the metric space $(\mathbb{R}^{|\mathcal{V}|}, \|\cdot\|_1)$. Consequently, graph diffusion can be construed as a composite of contraction maps. The PABI technique has been devised to privatize contractive iterations by injecting random noise per iteration. Empirical studies suggest that distributing noise throughout the diffusion steps can provide improved utility-privacy trade-offs compared to output perturbation alone [25]. This insight serves as a key motivation for employing PABI to establish privacy-preserving graph diffusion.

### 3.1 Preliminaries: Privacy Amplification by Iteration

The technique of Privacy Amplification by Iteration (PABI), originally introduced by Feldman et al. [26] for convex risk minimization problems via noisy gradient descent, bounds the privacy loss

of an iterative algorithm without releasing the full sequence of iterates. This approach applies to processes generated by Contractive Noisy Iteration (CNI) defined as follows.

**Definition 3** (Contractive Noisy Iteration (CNI) [26])**.** *Consider a Banach space $(\mathcal{X}, \|\cdot\|)$ with an initial random state $X_0 \in \mathcal{X}$, a series of contractions (i.e., c-Lipschitz functions, $c \leq 1$) $\psi_k : \mathcal{X} \to \mathcal{X}$, and a sequence of noise random variables $\{\xi_k\}$. Defining $\mathcal{B}$ as a convex bounded set, the Contractive Noisy Iteration $CNI(X_0, \{\psi_k\}, \{\xi_k\}, \mathcal{B})$ is governed by the update rule:*

$$X_{k+1} = \mathscr{P}_{\mathcal{B}}[\psi_t(X_k) + \xi_{k+1}] \tag{2}$$

*where $\mathscr{P}_{\mathcal{B}}$ is the projection operator onto $\mathcal{B}$, respecting the norm $\|\cdot\|$.*

In the PABI analysis by Feldman et al. [26], gradient descent is conceptualized as a contractive mapping $\psi_t$ in the $\ell_2$ space. Leveraging an additive Gaussian noise mechanism after each iteration, i.e., $\xi_k \sim \mathcal{N}(\mathbf{0}, \sigma^2 \mathbf{I})$, leads to the observation that the Rényi divergence of identical CNIs with differing initial conditions $X_0$ and $X_0'$ decays inversely with respect to the total number of iterations $K$. Specifically, it is observed that $\mathcal{D}_\alpha(X_K \| X_K') \leq \frac{\alpha \|\mathbf{X}_0 - \mathbf{X}_0'\|_2}{2K\sigma^2}$. Altschuler et al. [27] further extended this framework with improved bound as follows:

**Proposition 1.** *Let $X_K$ and $X_K'$ denote the outputs from $CNI(X_0, \{\psi_k\}, \{\xi_k\}, \mathcal{B})$ and $CNI(X_0, \{\psi_k'\}, \{\xi_k'\}, \mathcal{B})$, respectively, where $\xi_k, \xi_k' \sim \mathcal{N}(0, \sigma^2 I_d)$. Define distortion $\rho := \sup_{k,x} \|\psi_k(x) - \psi_k'(x)\|$ and let $\mathcal{B}$ have diameter $D$. If $\{\psi_k\}$ and $\{\psi_k'\}$ are contractions with coefficient $c < 1$, then for any $\tau \in \{0, \ldots, K-1\}$,*

$$\mathcal{D}_\alpha(X_K \| X_K') \leq \frac{\alpha}{\sigma^2} \Bigg[ \underbrace{(K-\tau)\rho^2}_{\text{Distortion Absorption}} + \underbrace{c^{2(K-\tau)}D^2}_{\text{PABI}} \Bigg]. \tag{3}$$

The bound in Eq. (3) demonstrates that the Rényi divergence between two CNIs can be quantified by the cumulative Rényi divergence over Gaussian noise with a distortion factor $\rho$ (the Distortion Absorption term), complemented by a PABI term. The latter indicates that identical contractive transformations applied to bounded processes reduce privacy leakage in an exponential manner. Note that the bound in Eq. (3) is convex with respect to $\tau$, optimized selection of $\tau$ leads to non-divergent upper bound $\tilde{\rho}(\ln(D^2/\tilde{\rho}) + 1)$ where $\tilde{\rho} = \rho^2/2\ln(1/c)$.

**Note on Parameter Set Diameter $D$.** The privacy bound in Eq. (3) relies on the assumption of bounded diameter $D$ of the parameter set $\mathcal{B}$ to upper bound $\infty$-Wasserstein distance (definition in App. A) between the coupled CNI processes $X_\tau$ and $X_\tau'$. Although in theory, the upper bound of Eq. (3) only depends on $\log D$ by optimizing $\tau$, we notice that the value $D$ is important to get a *practically meaningful* privacy bound. To tighten this term, we will introduce a novel $\infty$-Wasserstein distance tracking method that circumvents the need for the diameter parameter in Lemma 3 (detailed later): A high-level idea is to track the $\infty$-Wasserstein distance between noisy iterates via constructed couplings instead of using the default set diameter as an upper bound.

**Note on Noise Random Variables $\xi_k$.** The traditional PABI analysis primarily examines gradient descent within the $\ell_2$ space employing the Gaussian mechanism. In contrast, in the context of graph diffusions, modifications to Proposition 1 are necessary to accommodate the $\ell_1$ norm and the application of Laplace noise. This adaptation to the Laplace mechanism, crucial for the graph diffusion applications, has not been previously addressed in the literature to our knowledge.

### 3.2 Private Graph Diffusions

We now introduce our noisy graph diffusion framework, designed to ensure edge-level RDP and its personalized variant. Our approach consists of injecting Laplace noise into the contractive diffusion process and integrating a graph-dependent thresholding function to mitigate the high sensitivity associated with perturbations of low-degree nodes.

Given a graph diffusion process $\mathscr{D} = \{\phi_k\}_{k=1}^\infty$, we introduce a ***noisy graph diffusion*** $\mathscr{D}_{K,\sigma}$ where $K$ denotes the diffusion steps and $\sigma$ is the standard deviation of the added noise, constructed by a series of composing ***noisy graph diffusion mappings*** $\phi_{k,\sigma}$:

$$\mathscr{D}_{K,\sigma} = \phi_{K,\sigma} \circ \phi_{K-1,\sigma} \circ \cdots \circ \phi_{1,\sigma}, \text{ where } \phi_{k,\sigma}(\mathbf{s}_{k-1}) = \phi_k(f(\mathbf{s}_{k-1})) + \xi_k^{(1)} + \xi_k^{(2)}. \tag{4}$$

where $f$ is a *graph-dependent degree-based function* set as $f(\mathbf{x}) = \min(\max(\mathbf{x}, -\eta \cdot \mathbf{d}), \eta \cdot \mathbf{d})$ with a threshold parameter $\eta$ to balance privacy-utility trade-off. Specifically, $f$ clips the values of the diffusion vector according to node degrees. Notably, the thresholding function $f$ allows for negative signals, capturing scenarios where the diffusion coefficient $\gamma_{1,k}$ can be negative. Noise variables $\xi_k^{(1)}$ and $\xi_k^{(2)}$ are independently sampled from a Laplace distribution $\mathcal{L}(\mathbf{0}, \sigma)$. It is noteworthy that our framework can also be extended to accommodate Gaussian distributions. However, Gaussian noise has been shown to be suboptimal for graph diffusion in $\ell_1$ space, with empirical evidence provided in App. D.4.

**Design of Thresholding Function $f$.** In the noisy graph diffusion process, the role of the graph-dependent thresholding function $f$ is twofold. Firstly, $f$ ensures a bounded distance between the coupled diffusions over two adjacent graphs, analogous to the role of the general projection operator $\mathscr{P}_{\mathcal{B}}$ in the standard CNI as defined in Eq. (2).

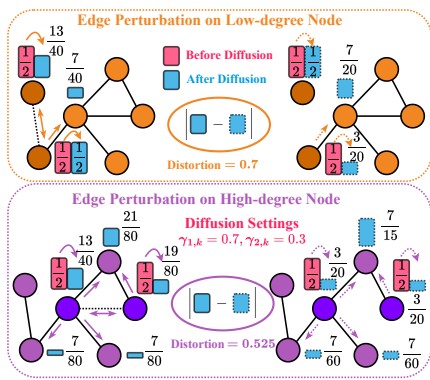

Figure 1: Illustration of Distortion from Edge Perturbations over Adjacent Graphs for Nodes with Low and High Degrees.

Such a bounding effect is also crucial for the later analysis of $\infty$-Wasserstein distance tracking in Lemma 3. Secondly, and more critically, our theoretical analysis reveals that edge perturbation affecting low-degree nodes results in increased distortion at each diffusion step (illustrated in Fig. 1). Uniform thresholding coupled with randomness injection for all nodes typically yields suboptimal performance in such cases. Our degree-dependent design naturally controls the distortion per iteration caused by low-degree nodes which helps with reducing the added noise. More detailed distortion analysis on $f$ is shown later in Lemma 2. The threshold parameter $\eta$ is commonly employed to optimize the privacy-utility trade-off in practical applications [20]. The empirical benefits of $f$ are explored in experiments detailed in Sec. 4.2.

**Discussion on Dual Noise Injection.** Our framework employs a noise-splitting technique, injecting dual Laplace noise at each diffusion step to construct non-divergent privacy bounds, as outlined in Eq. (3). Theoretical justifications for this design is provided in the proof sketch.

Following this, we present our main result on the privacy guarantee of noisy graph diffusion:

**Theorem 1** (Privacy Guarantees of Noisy Graph Diffusions). *Given a graph $\mathcal{G}$, an associate graph diffusion $\mathscr{D} = \{\phi_k\}_{k=1}^{\infty}$, then noisy graph diffusion mechanism $\mathscr{D}_{K,\sigma}$ ensures edge-level $(\alpha, \epsilon)$-RDP with $\epsilon$ satisfies:*

$$\epsilon \leq \min_{\tau \in \{0,1,\ldots,K-1\}} \left[ (K - \tau) \cdot g_\alpha(\sigma, \rho_{diff}) + g_\alpha\left( \sigma, \frac{\rho_{diff} \cdot (1 - \gamma_{max}^\tau)}{1 - \gamma_{max}} \cdot \gamma_{max}^{K-\tau} \right) \right] \quad (5)$$

*where $g_\alpha(\sigma, \rho) = \frac{1}{\alpha-1} \ln(\frac{\alpha}{2\alpha-1} \exp(\frac{\alpha-1}{\sigma}\rho) + \frac{\alpha-1}{2\alpha-1} \exp(-\frac{\alpha}{\sigma}\rho))$ denotes the Rényi divergence induced by the Laplace mechanism [44], and $\rho_{diff} = \max(4\gamma_{max}^{(1)}, 2\gamma_{max}) \cdot \eta$ represents the maximum single-step distortion incurred by diffusion on adjacent graphs that involves Lipschitz continuity coefficient $\gamma_{max}$, and maximum diffusion coefficient $\gamma_{max}^{(1)}$.*

*By selecting $\tau = \lceil K - \ln((\frac{1}{\rho_{diff}} + \frac{1}{1-\gamma_{max}}) \ln \frac{1}{\gamma_{max}}) / \ln(\frac{1}{\gamma_{max}}) \rceil$, privacy budget $\epsilon$ remains bounded by*

$$\epsilon \lesssim \frac{\rho_{diff}}{\sigma \cdot \ln\left(\frac{1}{\gamma_{max}}\right)} \left[ \ln\left( \left( \frac{1}{\rho_{diff}} + \frac{1}{1 - \gamma_{max}} \right) \ln \frac{1}{\gamma_{max}} \right) + 1 \right]. \quad (6)$$

The privacy bound in Eq. (5) consists of two components: the distortion absorption term (the first term on the RHS) and the PABI term (the second term in RHS). Distortion absorption quantifies the cumulative Rényi divergence over Laplace noise with single-step distortion $\rho_{\text{diff}}$, while the PABI term quantifies the exponential decay rate, echoing the result in Eq. (3). However, a key difference lies in our approach; instead of leveraging the projected set diameter $D$ to control the distance between coupled CNIs, our proposed $\infty$-Wasserstein tracking method yields a more practical term,

$\frac{\rho_{\text{diff}} \cdot (1 - \gamma_{\max}^{\tau})}{1 - \gamma_{\max}}$. Further details and utility evaluations of this tool are presented in the proof sketch and Sec. 4.2, respectively.

The function $g_\alpha(\sigma, \rho)$, which measures Rényi divergence for the Laplace mechanism, increases with distortion $\rho$ and decreases with noise scale $\sigma$. This behavior implies that reducing distortion and increasing the noise scale enhances privacy. To achieve better calibrated noise within a given privacy budget $\epsilon$, we calculate the two terms in Eq. (5) for each $\tau$. Leveraging the monotonicity of $g_\alpha(\sigma, \rho)$, we employ a binary search to identify the appropriate noise scale $\sigma$. The optimal noise scale is then determined by selecting the minimum value across various $\tau$ values, achieving this efficiently with linear complexity relative to $\tau$.

It is important to note that the maximum single-step distortion $\rho_{\text{diff}}$ in Eq. (5) is tight and conveys several messages. First, as defined in Eq. (1), when the diffusion process is relatively slow (i.e., $\gamma_{1,k} < \gamma_{2,k}$), the distortion remains tight, governed by the Lipschitz constant $\gamma_{\max}$ of the diffusion mapping. In contrast, when the diffusion is relatively fast (i.e., $\gamma_{1,k} \geq \gamma_{2,k}$), the distortion bound becomes asymptotically tight, depending on graph structures, with worst-case scenarios detailed in App. B.1.

In Eq. (6), we demonstrate the convergence of the privacy budget with respect to diffusion steps $K$. This approach differs from the adaptive composition theorem [42], which analyzes how privacy guarantees degrade when composed mechanisms are applied. Although this method has commonly been employed to protect privacy in graph learning models [52, 53, 56], it leads to a linear increase in the privacy budget with the number of iterations $K$ under Rényi divergence [44], potentially resulting in unbounded losses as $K$ grows to infinity. More importantly, even for a small number of diffusion steps, our framework achieves a significantly better privacy budget under practical PPR diffusion settings, as illustrated in Fig. 2. Further empirical evaluations are detailed in App. D.4.

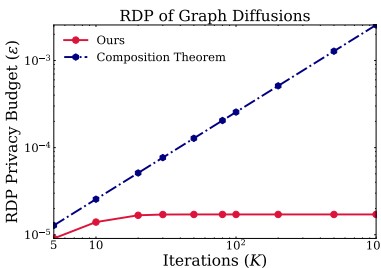

Figure 2: RDP vs. Total Diffusion Step $K$ with $\gamma_{1,k} = 0.8, \gamma_{2,k} = 0, \gamma_{3,k} = 0.2, \alpha = 2, \sigma = 0.01$, and $\eta = 10^{-5}$.

### 3.3 Proof Sketch of Theorem 1

**Proof Idea.** Similar to Eq. (3), the privacy loss of adjacent graph diffusion processes can be bounded as the sum of distortion absorption term incurred by Laplace noise and a PABI term at intermediate step $\tau$ (Step 1 & 2). Subsequently, we explore degree-based thresholding to manage distortion, achieving a superior utility-privacy tradeoff (Step 3), and introduce $\infty$-Wasserstein distance tracking to further tighten the divergence at $\tau$ (Step 4).

**Step 1: Interpretation of Iterates as Conditional CNI Sequences.** Consider the coupled graph diffusions $\mathscr{D} = \{\phi_k\}_{k=1}^\infty$ and $\mathscr{D}' = \{\phi'_k\}_{k=1}^\infty$, and the thresholding functions $f$ and $f'$, operating over adjacent graphs $\mathcal{G}$ and $\mathcal{G}'$, respectively. In each diffusion step, the first noise component constructs noisy iterates, while the second noise component is used to absorb distortion incurred between the adjacent graphs. We encapsulate the discussion as follows:

$$\mathbf{s}_k = \underbrace{\phi_k(f(\mathbf{s}_{k-1})) + \xi_k^{(1)}}_{\text{Identical CNI}} + \xi_k^{(2)}, \quad \mathbf{s}'_k = \phi'_k(f'(\mathbf{s}'_{k-1})) + \xi_k'^{(1)} + \xi_k'^{(2)} \stackrel{d}{=} \underbrace{\phi_k(f(\mathbf{s}'_{k-1})) + \xi_k'^{(1)}}_{\text{Identical CNI}} + \tilde{\xi}_k'^{(2)}.$$

where $\xi_k^{(1)}, \xi_k^{(2)}, \xi_k'^{(1)}, \xi_k'^{(2)} \sim \mathcal{L}(\mathbf{0}, \sigma)$, and $\tilde{\xi}_k'^{(2)} \sim \mathcal{L}\left(\phi'_k(f'(\mathbf{s}'_{k-1})) - \phi_k(f(\mathbf{s}'_{k-1})), \sigma\right)$, and $\stackrel{d}{=}$ denotes equality in distribution.

When the distortion $\phi'_k(f'(\mathbf{s}'_{k-1})) - \phi_k(f(\mathbf{s}'_{k-1}))$ is absorbed by the conditional event of noise variables, i.e., $\xi_k^{(2)} = \tilde{\xi}_k^{(2)}$, the coupled diffusion vectors evolve with identical CNIs through the contractive mapping $\phi_k \circ f$. Note that the Laplace distribution is essential for fully exploiting $\ell_1$ distortion in our analysis.

**Step 2: Bounding Privacy Loss through Distortion Absorption and PABI.** The privacy loss of coupled iterates $\mathcal{D}_\alpha(\mathbf{s}_K \| \mathbf{s}'_K)$ can be bounded by the distortion from graph diffusion, and PABI:

$$\mathcal{D}_\alpha(\mathbf{s}_K \| \mathbf{s}'_K) \leq \underbrace{\mathcal{D}_\alpha(\xi_{\tau+1:K}^{(2)} \| \tilde{\xi}_{\tau+1:K}'^{(2)})}_{\text{Distortion Absorption}} + \underbrace{\sup_\zeta \mathcal{D}_\alpha(\mathbf{s}_K | \xi_{\tau+1:K}^{(2)} = \zeta \| \mathbf{s}'_K | \tilde{\xi}_{\tau+1:K}'^{(2)} = \zeta)}_{\text{PABI}} \tag{7}$$

This inequality arises from leveraging the post-processing and strong composition rules of Rényi divergence. Here, $\zeta$ represents a joint noise realization, and the parameter $\tau$ is introduced to balance the privacy leakage from the two terms — the divergence between the shifted noise variables accumulated from step $\tau + 1$ to step $K$ (Distortion Absorption), and the divergence across conditional CNIs employing identical transformations $\phi_k \circ f$ (PABI).

**Step 3: Bounding Distortion Absorption**

**Lemma 2** (Absorption of Distortion in Laplace Distribution). *For any $\tau \in \{0, 1, ..., K-1\}$, we have*

$$\mathcal{D}_\alpha(\xi^{(2)}_{\tau+1:K} \| \tilde{\xi}'^{(2)}_{\tau+1:K}) \leq (K - \tau) g_\alpha(\sigma, \tilde{\rho}) \tag{8}$$

*Here, $\tilde{\rho}$ quantifies the maximum distortion introduced by a single-step diffusion and is determined by the thresholding function $f$ normalized by node degrees, i.e., $\frac{[f(\mathbf{s}_k)]_i}{d_i}$.*

The observation on $\tilde{\rho}$ highlights the importance of a degree-based design for the thresholding function $f$. Uniform thresholding across all nodes results in distortion proportional to $\frac{1}{d_{\min}}$, introducing unnecessarily large noise induced by low-degree nodes and degrading overall performance. This in principle inspires the choice of $f$ relying on node degrees. Consequently, $\tilde{\rho}$ is tightly bounded by $\rho_{\text{diff}} = \max(4\gamma^{(1)}_{\max}, 2\gamma_{\max}) \cdot \eta$.

**Step 4: Upper Bounding PABI with $\infty$-Wasserstein distance tracking.** To perform tight privacy analysis for the second term in Eq. (7), we develop a novel $\infty$-Wasserstein distance tracking method for coupled CNIs, where we denote the $\infty$-Wasserstein distance at step $\tau$ by $w_\tau$. This method discards the original boundedness condition in PABI (Eq. (3) Second Term), which relies on the diameter $D$.

**Lemma 3** (PABI with $\infty$-Wasserstein Distance Tracking). *Given two coupled graph diffusions mentioned above, for any $\tau \in \{0, 1, ..., K-1\}$, any noise realization $\zeta$, we have*

$$\mathcal{D}_\alpha(\mathbf{s}_K | \xi^{(2)}_{\tau+1:K} = \zeta \| \mathbf{s}'_K | \tilde{\xi}'^{(2)}_{\tau+1:K} = \zeta) \leq g_\alpha(\sigma, \gamma^{K-\tau}_{max} w_\tau) \tag{9}$$

Figure 3: **Setting:** Graph Diffusion with $\gamma_{1,k} = 0.8, \gamma_{2,k} = 0, \gamma_{3,k} = 0.2$.

*where the tracked $\infty$-Wasserstein distance over coupled CNIs is given by $w_\tau = \frac{\rho_{diff} \cdot (1 - \gamma^\tau_{max})}{1 - \gamma_{max}}$ and is naturally upper bounded by $\frac{\rho_{diff}}{1 - \gamma_{max}} := w$.*

We argue that using $w_\tau$ (or the upper bound $w$) instead of the default diameter $D$ is crucial to make the algorithm practically useful. There is no numerical evaluation in the previous study [27]. Numerical comparison between $w$ and the diameter of thresholding function $D = \eta \cdot \sum_{i=1}^{|\mathcal{V}|} d_i$, using the real-world *BlogCatalog* dataset (detailed in Sec. 4), is illustrated in Fig.3. $w$ achieves orders-of-magnitude improvement, which is still significant even if $D$ impacts privacy loss via a logarithmic term. Further empirical validations demonstrating significant utility improvements are detailed in Sec. 4.2.

By substituting the bounds from Eq. (8) and Eq. (9) into Eq. (7), we establish Theorem 1.

### 3.4 Personalized Graph Diffusion Algorithms with Application in PPR Diffusion

In practice, graph diffusions often originate from a single node $\mathbf{e}_i$, personalizing the algorithm to this seed node (user). Since the output is provided only to the seed node, protecting its edge connections (one-hop neighbors) becomes unnecessary, ensuring no privacy leakage in the first diffusion step under personalized privacy guarantees. Consequently, the thresholding function is tailored as follows: $f(\mathbf{x}) = \min(\max(\mathbf{x}, -\eta \cdot \tilde{\mathbf{d}}), \eta \cdot \tilde{\mathbf{d}})$, where $[\tilde{\mathbf{d}}]_j = [\mathbf{d}]_j$ for $j \neq i$ and $[\tilde{\mathbf{d}}]_i$ can be set to any positive threshold, i.e., no control is needed for the diffusion over seed node. We employ personalized edge-level RDP (Definition 2), caring two adjacent graphs with a difference only in a single edge not linked directly to the seed node. This approach is encapsulated in the following theorem:

**Theorem 4** (Privacy Guarantees for Personalized Noisy Graph Diffusions). *Given a graph $\mathcal{G}$, an associate graph diffusion $\mathscr{D} = \{\phi_k\}_{k=1}^\infty$, then personalized noisy graph diffusion mechanism $\mathscr{D}_{K,\sigma}$ with corresponding $f(\mathbf{x}) = \min(\max(\mathbf{x}, -\eta \cdot \tilde{\mathbf{d}}), \eta \cdot \tilde{\mathbf{d}})$ ensures personalized edge-level $(\alpha, \epsilon)$-RDP*

*with $\epsilon$ satisfies:*

$$\epsilon \leq \min_{\tau \in \{0,1,\dots,K-1\}} \left[ (K - \tau) \cdot g_\alpha \left( \sigma, \rho_{diff} \cdot \mathbb{1}_{\tau \neq 0} \right) + g_\alpha \left( \sigma, \frac{\rho_{diff} \cdot (1 - \gamma_{max}^\tau)}{1 - \gamma_{max}} \cdot \gamma_{max}^{K-\tau} \right) \right] \quad (10)$$

*where $\mathbb{1}$ denote indicator function.*

Note that a key difference from Theorem 1 is that in personalized privacy settings, there is no privacy leakage in the first diffusion step ($K = 1$).

**PPR Application.** Among various graph diffusions, PPR stands out as a prevalent node proximity metric extensively used in graph mining and network analysis. We may apply our noisy diffusion framework to PPR diffusion. We consider PPR with lazy random walk as follows:

$$\mathscr{D}(\mathbf{s}) = (1 - \beta) \sum_{k=0}^{\infty} \beta^k \mathbf{W}\mathbf{s} = \lim_{K \to \infty} \phi_K \circ \cdots \circ \phi_1(\mathbf{s}), \text{ where } \phi_k(\mathbf{x}) = \beta\mathbf{W}\mathbf{x} + (1 - \beta)\mathbf{s}. \quad (11)$$

where lazy random walk matrix $\mathbf{W} = \frac{1}{2}(\mathbf{P} + \mathbf{I})$ and $(1 - \beta)$ represents teleport probability with $\beta \in (0, 1]$.

Our framework incorporates noise into the diffusion process of each step of PPR. The privacy guarantees for this noisy PPR are derived from Theorem 4 with $\rho_{diff} = 2\beta\eta$ and $\gamma_{max} = \beta$. Note that, since $\gamma_{j,k} > 0$ for all $j \in \{1, 2, 3\}$ in PPR scenarios, all signals propagating among nodes should be non-negative. Consequently, the degree-based thresholding function $f$ can be further modified as $f(\mathbf{x}) = \min(\max(\mathbf{x}, \mathbf{0}), \eta \cdot \tilde{\mathbf{d}})$.

## 4 Experiments

In this section, we present empirical evaluations to support our theoretical findings. Specifically, we apply the widely-used PPR algorithm (Sec. 3.4) to real-world graphs. In practice, we also include a projection step onto the unit $\ell_1$ ball after injecting Laplace noise at each diffusion step. This adjustment has been observed to slightly improve the utility of the resulting PPR without impacting our theoretical analysis (see App. B.1 for details). We focus on the accuracy of noisy PPR in ranking tasks under personalized edge-level DP due to its practicality as noted in [20].

**Benchmark Datasets.** We conduct experiments on the following datasets: *BlogCatalog* [57], a social network of bloggers with 10,312 nodes and 333,983 edges; *Flickr* [57], a photo-sharing social network with 80,513 nodes and 5,899,882 edges; and *TheMarker* [58], an online social network with 69,400 nodes and 1,600,000 edges.

**Baselines.** Our experimental study includes two baselines. DP-PUSHFLOWCAP [20] is the only private PPR method using Laplace output perturbation, adapting the approximate PPR algorithm with push operations [41]. Edge-Flipping is the other baseline, which uses a randomized response mechanism [42] on the adjacency matrix, excluding seed node-connected edges in personalized scenarios. Entries are replaced with values in $\{0, 1\}$ uniformly at random with probability $p$ (detailed in App. E), or retained otherwise. This method requires $\mathcal{O}(|\mathcal{V}|^2)$ time to generate a private adjacency matrix and increases its edge density, which limits its practicality. Both our approach and DP-PUSHFLOWCAP offer better scalability. A comparison of running times between different approaches is provided in App. D.2. In all experiments, we only report results if a single trial can be completed within 12 hours on an AMD EPYC 7763 64-Core Processor, and thus Edge-Flipping cannot be run on *Flickr*.

**Metrics.** For utility, we employ two ranking-based metrics: normalized discounted cumulative gain at R (NDCG@R) and Recall@R [59], where R denotes the cutoff point for the top-ranked items in the list. In our experiments, R is set to 100. For privacy assessments, we utilize the personalized edge-level $(\epsilon, \delta)$-DP, with $\delta$ set to $\frac{1}{\#\text{edges}}$ following [53]. To ensure a fair comparison, both DP-PUSHFLOWCAP and Edge-Flipping are analyzed using RDP. The privacy budgets for all methods are subsequently converted to DP from RDP results, as elaborated in App. E. All results are reported as averages over 100 independent trials, with 95% confidence intervals.

### 4.1 Evaluating Privacy-Utility Tradeoffs on Ranking Tasks

In this series of experiments, we aim to assess the ranking performance of our noisy graph diffusion (as delineated in Sec. 3.4) compared with baselines on real-world graphs. We specifically examine privacy budget $\epsilon$ ranging from $10^{-2}$ to 1, PPR with parameter $\beta = 0.8$. Considering that both our

approach and `DP-PUSHFLOWCAP` employ a thresholding parameter $\eta$ to balance the privacy-utility trade-off, we select $\eta$ from a set of seven values spanning orders of magnitude from $10^{-10}$ to $10^{-4}$, a range empirically determined to be optimal across various datasets for both methods ($\eta = 10^{-6}$ is chosen for `DP-PUSHFLOWCAP` in [20]). For each experiment, we randomly choose an initial seed for diffusion and execute PPR for 100 iterations following [20]. We report the average NDCG@100 score and Recall@100 compared to the standard noise-free PPR diffusion (Eq. (11)) over 100 independent trials in Fig. 4 and Fig. 5 respectively.

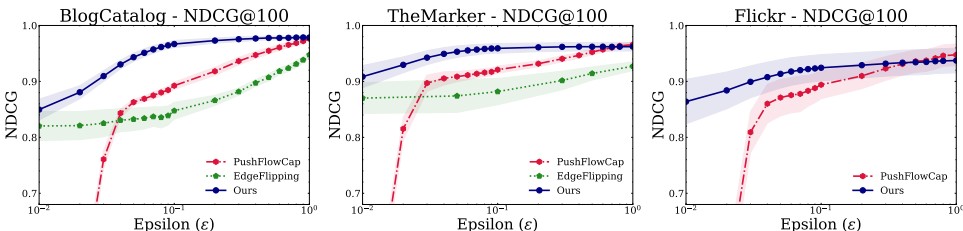

Figure 4: Trade-off between NDCG and Personalized Edge-level Privacy.

**Results for NDCG@100.** As illustrated in Fig. 4, our noisy graph diffusion surpasses both baselines across all three datasets, where values below 0.7 are ignored. In a strong privacy regime ($\epsilon \leq 0.5$), our approach demonstrates significant improvement over `DP-PUSHFLOWCAP`, which relies on output perturbation. This validates our claim that a noisy process achieves a superior privacy-utility trade-off in stringent privacy settings.

**Results for Recall@100.** Fig. 5 illustrates the overlap of the top-100 predictions of privacy-preserving PPR variants with standard PPR. Across all datasets, our method outperforms two baselines for $\epsilon$ values ranging from $10^{-2}$ to 1, further substantiating the advantages of our framework.

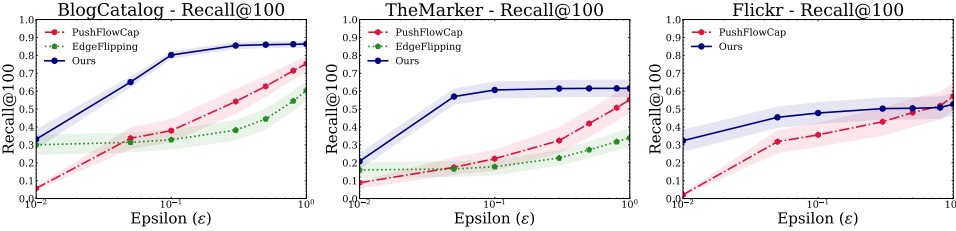

Figure 5: Trade-off between Recall and Personalized Edge-level Privacy.

Additional experiments on the sensitivity of ranking performance over variations in $\eta$ are deferred to App. D.3, which demonstrate that our approach is significantly more robust on the choice of the hyperparameter $\eta$.

### 4.2 Ablation Study

In this section, we conduct an ablation study to verify the effectiveness of our theory-guided designs, including the degree-based thresholding function $f$, the $\infty$-Wasserstein distance tracking tool. Experiments were conducted on BlogCatalog, utilizing PPR diffusion with parameter $\beta = 0.8$, and a total of $K = 100$ diffusion steps. Additional ablation studies focusing on variations in noise type and comparative analyses of noise scales across different methods are detailed in App. D.4.

**Degree-based Thresholding Function & $\infty$-Wasserstein Tracking.** We evaluate the effectiveness of our graph-dependent thresholding function $f$ and $\infty$-Wasserstein distance tracking. We vary $\eta$ uniformly across seven discrete values in from $10^{-4}$ to $10^{-10}$ and report the optimal performance for all methods in Fig. 6. Firstly, we compare the degree-based thresholding function $f(\mathbf{s}) = \min(\max(\mathbf{x}, 0), \eta \cdot \mathbf{d})$ (Red $\bullet$) against its graph-independent variant $\tilde{f}(\mathbf{s}) = \min(\max(\mathbf{x}, 0), \eta \cdot \mathbf{1})$ (Red $\times$), which uniformly clips over nodes regardless of their degree. As $\epsilon$ varies from 0.1 to 3, the degree-based thresholding function consistently outperforms its graph-independent counterpart, with a margin of 0.15 to 0.2 in the NDCG score.

This verifies the effectiveness of graph-dependent thresholding function $f$, which can balance the privacy-utility trade-off more effectively by focusing less on sensitive low-degree nodes.

Note that the graph-dependent (in)dependent thresholding function $f$ ($\tilde{f}$) naturally induces a diameter $\eta|\mathcal{V}|$ ($2\eta|\mathcal{E}|$), respectively, which can be utilized in the modification of Theorem 1 for privacy accounting. However, the graph dependent thresholding function $f$ results in a significantly larger diameter, which substantially degrades utility. This underscores the importance of leveraging $\infty$-Wasserstein tracking.

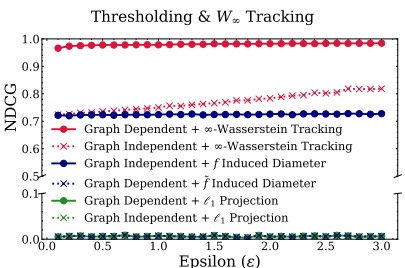

Figure 6: Effectiveness of degree-based thresholding function $f$ and $\infty$-Wasserstein distance tracking.

We showcase the benefits of our $\infty$-Wasserstein distance tracking analysis (Red Lines) on the privacy-utility tradeoff against diameter induced bounds derived from $\ell_1$ projection (Green Lines) or thresholding function-induced diameter (Blue Lines). The privacy bound of the $\ell_1$ projection is established based on a Laplace modification of Eq. (3) with a diameter of 1. As shown in Fig. 6, methods employing $\infty$-Wasserstein distance tracking analysis consistently outperform those based on $\ell_1$ projection and thresholding-induced diameter. We conclude that our $\infty$-Wasserstein tracking analysis and the design of graph-dependent thresholding function $f$ markedly enhance the privacy-utility tradeoff.

## 5 Conclusion

In summary, we introduce a noisy graph diffusion framework for edge-level privacy protection, utilizing a novel application of PABI in $\ell_1$ space. By incorporating a theory-guided design for a graph-dependent thresholding function and employing a new $\infty$-Wasserstein distance tracking tool, we outperform SOTA methods in ranking performance on benchmark graph datasets.

**Societal Impact and Limitations.** Our work advances the development of DP graph algorithms, offering strong privacy protection when properly implemented. For its limitations, we refer readers to standard textbooks on the subject [42]. The effectiveness of our noisy graph diffusion framework has been demonstrated primarily in the context of PPR diffusion. Extending these results to other types of graph diffusions, such as heat kernel diffusion, is a promising direction for future research. This paper focuses on edge-level privacy protections; exploring extensions to node-level DP protections could further broaden the scope of our research.

## Acknowledge

This work is supported by NSF awards CCF-2402816, IIS-2239565, and JPMC faculty award 2023. The authors would like to thank Haoyu Wang, Haoteng Yin for the valuable discussion. The authors also would like to thank Alessandro Epasto for the discussion about the implementation of the output-perturbation-based differentially private Personalized Pagerank.

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

# Appendix

## Contents

# A    Preliminaries

In this section, definitions and lemmas are introduced to facilitate the presentation of proofs. Recall that the contraction mapping denote a self-mapping $l$-Lipschitz continuous function with $l \in [0, 1]$, and the $\infty$-Wasserstein distance is defined as $W_\infty(\mu, \nu) = \inf_{\gamma \in \Gamma(\mu,\nu)} \mathrm{ess\,sup}_{(x,y)\sim\gamma} \|x - y\|$, where $\Gamma(\mu, \nu)$ represents the collection of all couplings between $\mu$ and $\nu$. Further, define $\mathcal{R}_\alpha(\sigma, \rho) := \sup_{\mathbf{r}:\|\mathbf{r}\|\leq\rho} \mathcal{D}_\alpha(\xi + \mathbf{r}\|\xi)$, where $\xi \sim \mathcal{L}(\mathbf{0}, \sigma)$, measures the extent to which the Laplacian distribution can absorb shifts of magnitude $\rho$. For clarity, $\mathcal{D}_\alpha(\mu\|\nu)$ and $\mathcal{D}_\alpha(X\|Y)$, with $X \sim \mu$ and $Y \sim \nu$, are used interchangeably when the context is clear. Besides, we denote $X_{i:j}, i, j \in \mathbb{Z}_+, i \leq j$ as joint couple of $(X_i, X_{i+1}, \ldots, X_j)$. We introduce Shifted Rényi Divergence as follows:

**Definition A.1** (Shifted Rényi Divergence). Let $\mu$ and $\nu$ be distributions defined over a Banach space $(\mathcal{X}, \|\cdot\|)$. For parameter $z \geq 0$ and $\alpha \geq 1$, the $z$-shifted Rényi divergence between $\mu$ and $\nu$ is defined as

$$\mathcal{D}_\alpha^{(z)}(\mu\|\nu) = \inf_{\mu':W_\infty(\mu,\mu')\leq z} \mathcal{D}_\alpha(\mu'\|\nu) \tag{12}$$

The following two lemmas illustrate how shifted Rényi divergence behaves under noise convolution and contraction mapping.

**Lemma A.1** (Shift-reduction lemma [26]). Let $\mu, \nu, \varrho$ be probability distributions on $\mathbb{R}^n$. For any $\rho \geq 0$,

$$\mathcal{D}_\alpha^{(z)}(\mu * \varrho \| \nu * \varrho) \leq \mathcal{D}_\alpha^{(z+\rho)}(\mu\|\nu) + \sup_{\mathbf{r}:\|\mathbf{r}\|\leq\rho} \mathcal{D}_\alpha(\varrho + \mathbf{r}\|\varrho). \tag{13}$$

**Lemma A.2** (Contraction-reduction lemma [27]). Suppose $\psi, \psi'$ are random functions from $\mathbb{R}^n$ to $\mathbb{R}^n$ such that (i) each is a strict $c$-contraction almost surely, and (ii) there exists a coupling of $(\psi, \psi')$ under which $\sup_{\mathbf{x}} \|\psi(\mathbf{x}) - \psi'(\mathbf{x})\| \leq \rho$ almost surely. Then for any probability distributions $\mu$ and $\nu$ on $\mathbb{R}^n$,

$$\mathcal{D}_\alpha^{(cz+\rho)}(\psi_\#\mu\|\psi'_\#\nu) \leq \mathcal{D}_\alpha^{(z)}(\mu\|\nu). \tag{14}$$

Further, we recall two properties of RDP in our analysis.

**Lemma A.3** (Post-processing Property of Rényi Divergence). For any Rényi parameter $\alpha \geq 1$, any random function $f$, and any probability measures $\mu, \nu$, we have

$$\mathcal{D}_\alpha(f_\#\mu\|f_\#\nu) \leq \mathcal{D}_\alpha(\mu\|\nu) \tag{15}$$

**Lemma A.4** (Strong composition for RDP). For any Rényi parameter $\alpha \geq 1$, and any two sequences of random variables $X_1, \ldots, X_k$ and $Y_1, \ldots, Y_k$,

$$\mathcal{D}_\alpha(\mathbb{P}_{X_{1:k}}\|\mathbb{P}_{Y_{1:k}}) \leq \sum_{i=1}^{k} \sup \mathcal{D}_\alpha(\mathbb{P}_{X_i|X_{i-1}=x_{i-1}}\|\mathbb{P}_{Y_i|Y_{i-1}=x_{i-1}}).$$

# B    Privacy Guarantees for Noisy Graph Diffusion: Analytical Proofs

In this section, we present the main proof for the results in Theorem 1 as detailed in Sec. 3.

## B.1    Main Proof

*Proof of Theorem 1.* Given a seed $\mathbf{s}$, a total diffusion step $K$, a noise scale $\sigma$, and degree-based thresholding functions $f, f'$, we consider two coupled graph diffusions $\mathscr{D}_{K,\sigma}(\mathbf{s}), \mathscr{D}'_{K,\sigma}(\mathbf{s})$ which propogate on two joint edge-level adjacent graphs $\mathcal{G}, \mathcal{G}'$ respectively with diffusion mapping $\phi_k, \phi'_k, k \in [K]$. Specifically, we have

$$\mathscr{D}_{K,\sigma}(\mathbf{s}): \quad \mathbf{s}_k = \phi_k(f(\mathbf{s}_{k-1})) + \xi_k^{(1)} + \xi_k^{(2)}, 1 \leq k \leq K, \tag{16}$$

$$\mathscr{D}'_{K,\sigma}(\mathbf{s}): \quad \mathbf{s}'_k = \phi'_k(f'(\mathbf{s}'_{k-1})) + \xi_k^{'(1)} + \xi_k^{'(2)}, 1 \leq k \leq K. \tag{17}$$

$$\tag{18}$$

where $\xi_k^{(1)}, \xi_k'^{(1)}, \xi_k^{(2)}, \xi_k'^{(2)}$ are distributed according to the law $\mathcal{L}(\mathbf{0}, \sigma)$.

**Step 1: Distortion of Single-Step Graph Diffusion** In each diffusion step, a distortion over the diffusion mappings $\phi_k \circ f$ is introduced due to the graph adjacency between the graph diffusions $\mathscr{D}_{K,\sigma}(\mathbf{s})$ and $\mathscr{D}'_{K,\sigma}(\mathbf{s})$. This distortion, which is crucial for the subsequent analysis, is characterized by the following lemma:

**Lemma B.1** (Distortion of Graph Diffusion). Given two graph diffusions $\mathscr{D}_{K,\sigma}(\mathbf{s}), \mathscr{D}'_{K,\sigma}(\mathbf{s})$ mentioned above. Let $f, f'$ denote corresponding graph-dependent degree-based thresholding operators, i.e., $f(\mathbf{x}) = \min(\max(\mathbf{x}, -\eta \cdot \mathbf{d}), \eta \cdot \mathbf{d})$. The diffusion distortion satisfies:

$$\sup_{\mathbf{x} \in \mathbb{R}^n} \|\phi_k(f(\mathbf{x})) - \phi'_k(f'(\mathbf{x}))\|_1 \leq \max(4\gamma_{\max}^{(1)}, 2\gamma_{\max}) \cdot \eta \tag{19}$$

where Lipschitz constant $\gamma_{\max} = \max_k |\gamma_{1,k}| + |\gamma_{2,k}|$, and maximum diffusion coefficient $\gamma_{\max}^{(1)} = \max_k |\gamma_{1,k}|$.

We denote this single-step graph diffusion distortion bound as $\rho_{\text{diff}}$, i.e., $\rho_{\text{diff}} = \max(4\gamma_{\max}^{(1)}, 2\gamma_{\max}) \cdot \eta$. Moreover, incorporating an $\ell_1$-ball projection $\mathscr{P}_{\mathcal{B}}$ into the pipeline, i.e., adopting $\phi_k \circ f \circ \mathscr{P}_{\mathcal{B}}$, does not alter this distortion bound.

According to the proof, this distortion analysis is tight and conveys several key insights. First, when the graph diffusion process diffuses relatively slow, i.e., $\gamma_{1,k} < \gamma_{2,k}$, the distortion is tight and primarily governed by the Lipschitz constant of the diffusion mapping. In contrast, when the diffusion is relatively fast, the distortion is controlled by the maximum diffusion coefficient $\gamma_{1,k}$. In this latter scenario, the bound becomes asymptotically tight when the graph structure satisfies the condition that the nodes connected by the perturbed edge have no common neighbors, and their degrees tend to infinity. We raise a double star graph as an toy example that satisfies the above no common neighbor condition (see Fig. 7).

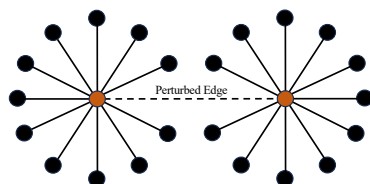

Figure 7: Double Star Graph.

**Step 3: Bounding the Privacy Loss via Shift Absorption and PABI.** Next, we consider the noisy diffusion process, where Laplacian noise is introduced during graph propagation. Note that, we focus on injecting noise at the initial step of the diffusion process, although the analysis can be directly extended to include noise injection at intermediate steps. For $k \geq 1$, drawing upon the noise-splitting approach outlined in [27], we mitigate the shifts caused by diffusion and thresholding by integrating these factors into the noise distribution. Consequently, we construct conditional CNI sequences with identical diffusion mappings for both processes. Specifically, we reformulate the diffusion process $\mathscr{D}'_{K,\Pi}(\mathbf{s})$ as follows:

$$\mathbf{s}'_k = \phi_k(f(\mathbf{s}'_{k-1})) + \xi_k'^{(1)} + \tilde{\xi}_k'^{(2)}, \text{ where } \tilde{\xi}'_k \sim \mathcal{L}(\phi'_k(f'(\mathbf{s}'_{k-1})) - \phi_k(f(\mathbf{s}'_{k-1})), \sigma_k) \tag{20}$$

where we introduce $\tilde{\xi}_k'^{(2)}$ as shifted Laplacian noise and noise scale $\sigma_k = \sigma$. Therefore, the coupled graph diffusion processes can be summarized as follows:

$$\mathbf{s}_k = \underbrace{\phi_k(f(\mathbf{s}_{k-1})) + \xi_k^{(1)}}_{\text{Identical CNI}} + \xi_k^{(2)}, \ \mathbf{s}'_k = \underbrace{\phi_k(f(\mathbf{s}'_{k-1})) + \xi_k'^{(1)}}_{\text{Identical CNI}} + \tilde{\xi}_k'^{(2)}, \ \forall k \geq 1. \tag{21}$$

Note that our objective is to establish an upper bound for $\mathcal{D}_\alpha(\mathbf{s}_K \| \mathbf{s}'_K)$, we claim that for any parameter $\tau \in \{0, 1, ..., K-1\}$,

$$\mathcal{D}_\alpha(\mathbf{s}_K \| \mathbf{s}'_K) \overset{(a)}{\leq} \mathcal{D}_\alpha(\mathbf{s}_K, \xi_{\tau+1:K}^{(2)} \| \mathbf{s}'_K, \tilde{\xi}_{\tau+1:K}'^{(2)}) \tag{22}$$

$$\overset{(b)}{\leq} \underbrace{\mathcal{D}_\alpha(\xi_{\tau+1:K}^{(2)} \| \tilde{\xi}_{\tau+1:K}'^{(2)})}_{\text{Shift Absorption}} + \underbrace{\sup_{\zeta_{\tau+1:K}} \mathcal{D}_\alpha(\mathbf{s}_K | \xi_{\tau+1:K}^{(2)} = \zeta_{\tau+1:K} \| \mathbf{s}'_K | \tilde{\xi}_{\tau+1:K}'^{(2)} = \zeta_{\tau+1:K})}_{\text{PABI}}$$

$$\tag{23}$$

where $\zeta_{\tau+1:K}$ is a noise realization, $(a)$ is by the post-processing inequality of Rényi Divergence (Lemma A.3), and $(b)$ is from the strong composition rule for Rényi divergence (Lemma A.4).

As demonstrated in Eq. 23, the privacy leakage can be upper bounded by the Rényi divergence across joint Laplacian distributions with a shift (shift absorption term) and the divergence across conditional CNIs employing identical transformations $\phi_k \circ f$ (PABI term). The parameter $\tau$ is introduced to balance the privacy leakage from shifts between noise distributions against the leakage from CNIs starting from different initial conditions. Both terms can be further bounded as detailed in the following lemmas. It is important to emphasize that, for the latter, we leverage $\infty$-Wasserstein distance tracking method that get rid of the diameter requirement in original PABI analysis. Besides, for each lemma below, we state that considering projection onto the $\ell_1$ ball does not affect the conclusions.

**(1) Upper bounding shift absorption term:**

**Lemma B.2** (Absorption of Shift in Laplacian Distribution). Given two coupled graph diffusions mentioned above, for any $\tau \geq 0$, the shift absorption term can be upper bounded by

$$\mathcal{D}_\alpha(\xi_{\tau+1:K}^{(2)} \| \tilde{\xi}_{\tau+1:K}'^{(2)}) \leq \sum_{k=\tau+1}^{K} g_\alpha(\sigma_k, \rho_{\text{diff}}) \tag{24}$$

where distortion $\rho_{\text{diff}} = \max(4\gamma_{\text{max}}^{(1)}, 2\gamma_{\text{max}}) \cdot \eta$, and shifted Laplace function $g_\alpha(\sigma, \rho) = \frac{1}{\alpha-1} \ln(\frac{\alpha}{2\alpha-1} \exp(\frac{\alpha-1}{\sigma}\rho) + \frac{\alpha-1}{2\alpha-1} \exp(-\frac{\alpha}{\sigma}\rho))$.

**(2) Upper bounding PABI term:**

**Lemma B.3** (PABI with Laplacian Distribution). Given two coupled graph diffusions mentioned above, for any $\tau \geq 0$, we have

$$\sup_{\zeta_{\tau+1:K}} \mathcal{D}_\alpha(\mathbf{s}_K | \xi_{\tau+1:K}^{(2)} = \zeta_{\tau+1:K} \| \mathbf{s}_K' | \tilde{\xi}_{\tau+1:K}'^{(2)} = \zeta_{\tau+1:K}) \leq \mathcal{D}_\alpha^{(w_\tau)}(\mathbf{s}_\tau \| \mathbf{s}_\tau') + g_\alpha(\sigma_K, \gamma_{\text{max}}^{K-\tau} w_\tau) \tag{25}$$

The lemma demonstrates that the PABI term is controlled by the shifted Rényi divergence at step $\tau$ and a privacy amplification term that decays exponentially at a rate of $\gamma_{\text{max}}^{K-\tau} w_\tau$. In previous analyses of the PABI for the Gaussian mechanism, [27] utilized the diameter of the bounded parameter set to further constrain the shift $w_\tau$. However, we have found that this bound does not satisfactorily achieve a favorable privacy-utility trade-off in practice. Subsequently, we develop a $\infty$-Wasserstein distance tracking method to more effectively bound this PABI term, offering a more practical approach.

**Lemma B.4** (PABI with $\infty$-Wasserstein Distance Tracking). Given two coupled graph diffusions mentioned above, for any $\tau \geq 0$, we have

$$W_\infty(\mathbf{s}_\tau \| \mathbf{s}_\tau') \leq \frac{\rho_{\text{diff}} \cdot (1 - \gamma_{\text{max}}^\tau)}{1 - \gamma_{\text{max}}} = w_\tau, \ \mathcal{D}_\alpha^{(w_\tau)}(\mathbf{s}_\tau \| \mathbf{s}_\tau') = 0 \tag{26}$$

where distortion $\rho_{\text{diff}} = \max(4\gamma_{\text{max}}^{(1)}, 2\gamma_{\text{max}}) \cdot \eta$.

By summarizing the above results (Lemma B.2, B.3, and B.4), we conclude the final results:

$$\mathcal{D}_\alpha(\mathbf{s}_K \| \mathbf{s}_K') \leq \mathcal{D}_\alpha(\xi_{\tau+1:K}^{(2)} \| \tilde{\xi}_{\tau+1:K}'^{(2)}) + \sup_{\zeta_{\tau+1:K}} \mathcal{D}_\alpha(\mathbf{s}_K | \xi_{\tau+1:K}^{(2)} = \zeta_{\tau+1:K} \| \mathbf{s}_K' | \tilde{\xi}_{\tau+1:K}'^{(2)} = \zeta_{\tau+1:K})$$

$$\leq \sum_{k=\tau+1}^{K} g_\alpha(\sigma_k, \rho_{\text{diff}}) + g_\alpha(\sigma_K, \gamma_{\text{max}}^{K-\tau} w_\tau) \tag{27}$$

$$= (K - \tau) \cdot g_\alpha(\sigma, \rho_{\text{diff}}) + g_\alpha(\sigma, \gamma_{\text{max}}^{K-\tau} \cdot \frac{\rho_{\text{diff}} \cdot (1 - \gamma_{\text{max}}^\tau)}{1 - \gamma_{\text{max}}}) \tag{28}$$

$\square$

## C   Discussion on Personalized Graph Diffusion

As discussed in Sec. 3.4, in many personalized scenarios, the definition of personalized edge-level privacy shifts to protect edges that are not incident to the source node. This adjustment offers additional benefits in distortion analysis (Lemma B.1): no distortion occurs in the first step. According to the proof of Lemma B.1, given a seed node $s$ with the corresponding initial vector $\mathbf{s}$, and letting $(u, v)$ denote the edge differing on adjacent graphs $\mathcal{G}$ and $\mathcal{G}'$ where $s \notin \{u, v\}$, we have:

$$\|(\mathbf{A}\mathbf{D}^{-1} - \mathbf{A}'\mathbf{D}'^{-1})\mathbf{s}\|_1 \leq 2 \left( \frac{[\mathbf{s}]_u}{d_u} + \frac{[\mathbf{s}]_v}{d_v} \right) = 0 \tag{29}$$

since $[\mathbf{s}]_u = [\mathbf{s}]_v = 0$. Thus, when analyzing noisy graph diffusion in a personalized scenario, this distinction results in a better bound for Lemma B.2. Further, Eq. (29) demonstrate that we can relax the thresholding over seed node to seek for better privacy-utility trade-offs. Therefore, for uniform noise scheduling, we obtain the corresponding results in Theorem 4:

$$\epsilon = \min_{\tau \in \{0,...,K-1\}} \left[ g_\alpha(\sigma, \frac{\rho_{\text{diff}} \cdot (1 - \gamma_{\text{max}}^\tau) \cdot \gamma_{\text{max}}^{K-\tau}}{1 - \gamma_{\text{max}}}) + (K - \tau) \cdot g_\alpha(\sigma, \rho_{\text{diff}} \cdot \mathbb{1}_{\tau \neq 0}) \right] \tag{30}$$

## D   Additional Experiments

### D.1   Datasets and Experimental Environment

As mentioned in Section 4, this paper includes three benchmark datasets: BlogCatalog, TheMarker, and Flickr. Their details are included in Table 1.

| Dataset | Size | | | Statistics | |
|---|---|---|---|---|---|
| | Nodes $|\mathcal{V}|$ | Edges $|\mathcal{E}|$ | Classes $|\mathcal{C}|$ | Avg. Deg. | Density |
| Blogcatalog | 10k | 334k | 39 | 64.8 | $6.3 \times 10^{-3}$ |
| TheMarker | 69.4k | 1.6M | NA | 47 | $6.8 \times 10^{-4}$ |
| Flickr | 80k | 5.8M | 195 | 146 | $1.5 \times 10^{-3}$ |

Table 1: Benchmark datasets and their statistics.

Experiments were performed on a server with two AMD EPYC 7763 64-Core Processors, 2TB DRAM, six NVIDIA RTX A6000 GPUs (each with 48GB of memory).

### D.2   Running Time

We report the running time for a single trial of each method across all datasets in Fig. 8. As illustrated in the figure, all single trial experiments of our method and `DP-PUSHFLOWCAP` are completed within 1 minute. In contrast, the edgeflipping mechanism exhibits significantly longer running times, ranging from 16 minutes to over 12 hours as the size of the graph increases.

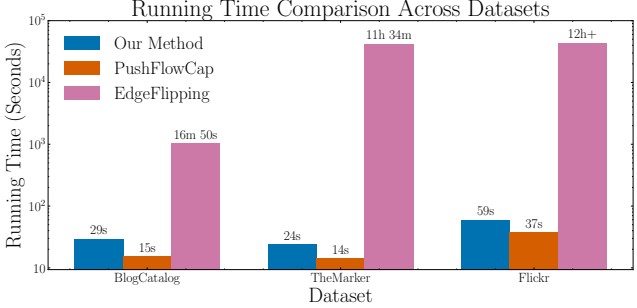

Figure 8: Running Time for a Single Trial of Experiments with a Privacy Budget of $\epsilon = 0.1$.

## D.3 Sensitivity of Ranking Performance to Variations in Threshold Parameter $\eta$.

In this series of experiments, we further investigate the sensitivity of the NDCG ranking performance with respect to the selection of $\eta$. Specifically, we enhance the granularity of the $\eta$ values within the existing range from $1 \times 10^{-10}$ to $1 \times 10^{-4}$ by selecting 20 equidistant points for each dataset within this interval. The NDCG ranking performance is depicted in Fig. 9. The left column represents the transition curve of our method, the middle column denotes the performance of DP-PUSHFLOWCAP, and the right column illustrates the performance gap between the two methods. For each privacy budget $\epsilon$, we highlight the optimal $\eta$ corresponding to the best performance (yellow for our method, cyan for DP-PUSHFLOWCAP). As demonstrated in the right column, when $\eta$ varies from $1 \times 10^{-10}$ to $1 \times 10^{-4}$, our method consistently outperforms DP-PUSHFLOWCAP up to $1 \times 10^{-6}$. This indicates the superior stability of hyperparameter selection for our method in terms of ranking performance.

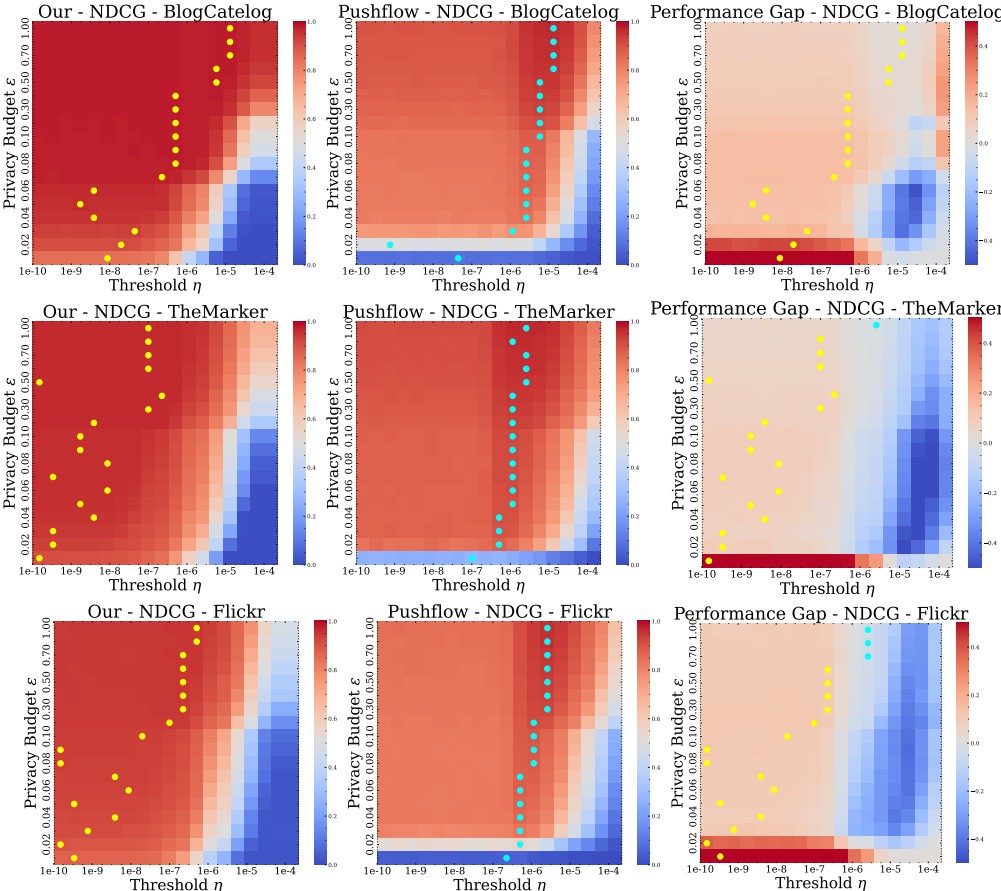

Figure 9: Transition Curves of Thresholding Parameter $\eta$ Relative to Privacy Budget $\epsilon$ for Three Benchmark Datasets.

## D.4 Ablation Study

In this section, we conduct additional ablation studies to compare the noise distributions, and noise schedules between our Theorem 1 and the Composition Theorem. Following the experimental settings outlined in Sec. 4.2, we perform experiments on the BlogCatalog dataset, utilizing PPR diffusion with a parameter $\beta = 0.8$ and a total of $K = 100$ diffusion steps.

**Noise Distributions: Laplace versus Gaussian.** We investigate the effectiveness of injecting Laplace noise as compared to Gaussian noise within our analysis framework. Recall that standard PABI analysis has primarily focused on the case of Gaussian noise. We report the ranking performance in Fig. 10 (Left). Within a strong privacy regime where $\epsilon$ ranges from 0.1 to 1, the performance gap

between the two noise types is approximately 0.05. This finding supports the superiority of Laplace noise injection for graph diffusion processes in $\ell_1$ space and is of practical importance.

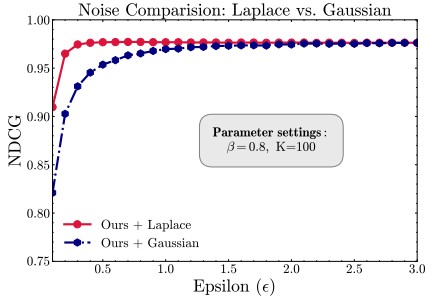 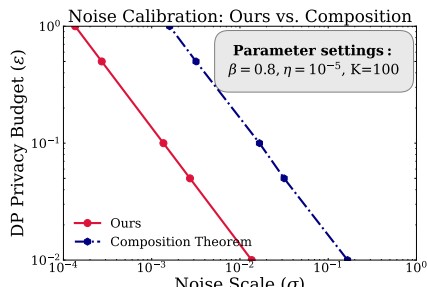

Figure 10: **Left:** Ranking performance with Laplace and Gaussian noise injection. **Right:** Comparison of noise scales between our method (Theorem 1) and the composition theorem under the DP metric.

**Theorem 1 versus Composition Theorem.** As we discussed in Section 3.2, one can naively adopt the DP composition theorem [43, 44] to establish the privacy guarantee for the same noisy graph diffusion. However, it is a general approach and does not take the contraction properties of graph diffusion into account, which leads to a worse privacy bound in our case. We examine the calibrated noise scales $\sigma$ under edge-level DP, converted from RDP. Figure 10 (Right) compares noise scales derived from our Theorem 1 (Red Line) against those from the standard RDP composition theorem (Blue Line). Our method achieves a noise scale that is 10 times smaller than that required by the composition theorem.

# E RDP to DP Conversion

First, we present the standard results on converting RDP to DP:

**Proposition E.1** (Conversion from RDP to DP [44]). *If $\mathcal{M}$ is an $(\alpha, \epsilon_{RDP})$-RDP mechanism, then it also satisfies $(\epsilon_{DP}, \delta)$-differential privacy, where $\epsilon_{DP} = \epsilon_{RDP} + \frac{\log \frac{1}{\delta}}{\alpha - 1}$ for any $\delta \in (0, 1)$.*

To determine the DP guarantees of the three mechanisms, we first calculate the RDP for each mechanism. Specifically, we use Theorem 4 for our method, sensitivity analysis in Theorem 4.3 from [20] with the dimensional Laplace mechanism under Rényi divergence (Eq. (46)) for DP-PUSHFLOWCAP, and Proposition 5 from [44] for EdgeFlipping. For all three mechanisms, the expression $\epsilon_{\text{RDP}} + \frac{\log \frac{1}{\delta}}{\alpha - 1}$ is convex in $\alpha$. By setting $\delta = \frac{1}{\#\text{edges}}$, we then search for the optimal $\alpha$ to minimize this expression and obtain $\epsilon_{\text{DP}}$.

# F Proof of Lemmas

## F.1 Proof of Lemma B.1

Given any $\mathbf{x} \in \mathbb{R}^n$, without loss of generality, we assume that the edge sets of two graphs satisfy $\mathcal{E} = \mathcal{E}' \cup (1, 2)$, i.e., we remove edge $(1, 2)$ from graph $\mathcal{G}$, resulting in graph $\mathcal{G}'$. Furthermore, let $\mathcal{N}(i)$ denote the neighbors of node $i$, and define the following index sets: $A := \mathcal{N}(1)\backslash\{2\}$, $B := \mathcal{N}(2)\backslash\{1\}$, $C := A \cap B$, $A' := A\backslash C$, and $B' := B\backslash C$. Additionally, we can rearrange the node indices in $\mathcal{G}$ such that $A' = \{3, \dots, |A'| + 2\}$, $B' = \{|A'| + 3, \dots, |A'| + |B'| + 2\}$, and $C = \{|A'| + |B'| + 3, \dots, |A'| + |B'| + |C| + 2\}$. We adopt the shorthand notation $f_i = [f(\mathbf{x})]_i$ and $\Delta f_i = [f(\mathbf{x})]_i - [f'(\mathbf{x})]_i$. We first consider the case where $d_1, d_2 \geq 2$, and define $m_1 := -\frac{f_1}{d_1(d_1-1)}, m_2 := -\frac{f_2}{d_2(d_2-1)}, \Delta m_1 := \frac{\Delta f_1}{d_1-1}, \Delta m_2 := \frac{\Delta f_2}{d_2-1}$. Since degree-based thresholding function $f$ is symmetric, WLOG, we can assume $f$ is non-negative, i.e., $f(\mathbf{x}) = \min(\max(\mathbf{x}, \mathbf{0}), \eta \cdot \mathbf{d})$. We have the following

$$\|\phi_k(f(\mathbf{x})) - \phi_k'(f'(\mathbf{x}))\|_1 = \|\phi_k(f(\mathbf{x})) - \phi_k'(f(\mathbf{x})) + \phi_k'(f(\mathbf{x})) - \phi_k'(f'(\mathbf{x}))\|_1 \tag{31}$$
$$= \|\gamma_{1,k}(\mathbf{P} - \mathbf{P}')f(\mathbf{x}) + (\gamma_{1,k}\mathbf{P}' + \gamma_{2,k}\mathbf{I})(f(\mathbf{x}) - f'(\mathbf{x}))\|_1 \tag{32}$$

$$=\|\gamma_{1,k}(\mathbf{A}\mathbf{D}^{-1}-\mathbf{A}'\mathbf{D}'^{-1})f(\mathbf{x})+(\gamma_{1,k}\mathbf{A}'\mathbf{D}'^{-1}+\gamma_{2,k}\mathbf{I})(f(\mathbf{x})-f'(\mathbf{x}))\|_1 \tag{33}$$

$$=\left\|\gamma_{1,k}\begin{bmatrix} 0 & \frac{1}{d_2} & 0 & \cdots & 0 \\ \frac{1}{d_1} & 0 & 0 & \cdots & 0 \\ \cdots & \cdots & \cdots & \cdots & \cdots \\ a_{n1}(\frac{1}{d_1}-\frac{1}{d_1-1}) & a_{n2}(\frac{1}{d_2}-\frac{1}{d_2-1}) & 0 & \cdots & 0 \end{bmatrix}\begin{bmatrix} f_1 \\ f_2 \\ \cdots \\ f_n \end{bmatrix}\right.$$

$$+(\gamma_{1,k}\mathbf{A}'\mathbf{D}'^{-1}+\gamma_{2,k})\begin{bmatrix} \Delta f_1 \\ \Delta f_2 \\ \cdots \\ \Delta f_n \end{bmatrix}\Bigg\|_1 \tag{34}$$

$$=\|\gamma_{1,k}\cdot[\frac{f_2}{d_2},\frac{f_1}{d_1},\underbrace{m_1,\cdots,m_1}_{|A'| \text{ Times}},\underbrace{m_2,\cdots,m_2}_{|B'| \text{ Times}},\underbrace{m_1+m_2,\cdots,m_1+m_2}_{|C| \text{ Times}},0,...,0]^T$$

$$+\gamma_{1,k}\cdot[0,0,\underbrace{\Delta m_1,...,\Delta m_1}_{|A'| \text{ Times}},\underbrace{\Delta m_2,...,\Delta m_2}_{|B'| \text{ Times}},\underbrace{\Delta m_1+\Delta m_2,...,\Delta m_1+\Delta m_2}_{|C| \text{ Times}}]^T+\gamma_{2,k}\cdot[\Delta f_1,\Delta f_1,0,...,0]^T\|_1 \tag{35}$$

$$\leq|\gamma_{1,k}|\cdot(|\frac{f_1}{d_1}|+|\frac{f_2}{d_2}|)+|\gamma_{1,k}|\cdot|A'|\cdot|(\frac{1}{d_1}-\frac{1}{d_1-1})f_1+\frac{\Delta f_1}{d_1-1}|+|\gamma_{1,k}|\cdot|B'|\cdot|(\frac{1}{d_2}-\frac{1}{d_2-1})f_2+\frac{\Delta f_2}{d_2-1}|$$

$$+|\gamma_{1,k}|\cdot|C|\cdot|(\frac{1}{d_1}-\frac{1}{d_1-1})f_1+\frac{\Delta f_1}{d_1-1}+(\frac{1}{d_2}-\frac{1}{d_2-1})f_2+\frac{\Delta f_2}{d_2-1}|+|\gamma_{2,k}|(|\Delta f_1|+|\Delta f_2|) \tag{36}$$

$$\overset{(a)}{=}|\gamma_{1,k}|\cdot(\frac{f_1}{d_1}+\frac{f_2}{d_2})+|\gamma_{1,k}|\cdot|A'|\cdot\frac{|d_1\Delta f_1-f_1|}{d_1(d_1-1)}+|\gamma_{1,k}|\cdot|B'|\cdot\frac{|d_2\Delta f_2-f_2|}{d_2(d_2-1)}$$

$$+|\gamma_{1,k}|\cdot|C|\cdot|\frac{d_1\Delta f_1-f_1}{d_1(d_1-1)}+\frac{d_2\Delta f_2-f_2}{d_2(d_2-1)}|+|\gamma_{2,k}|\cdot(\Delta f_1+\Delta f_2) \tag{37}$$

$$\overset{(b)}{\leq}|\gamma_{1,k}|\cdot(\frac{f_1}{d_1}+\frac{f_2}{d_2})+|\gamma_{1,k}|\cdot\frac{|d_1\Delta f_1-f_1|}{d_1}+|\gamma_{1,k}|\cdot\frac{|d_2\Delta f_2-f_2|}{d_2}+|\gamma_{2,k}|\cdot(\Delta f_1+\Delta f_2) \tag{38}$$

$$\overset{(c)}{=}|\gamma_{1,k}|\cdot(\frac{f_1}{d_1}+\frac{f_2}{d_2})+|\gamma_{1,k}|\cdot\frac{f_1-d_1\Delta f_1}{d_1}+|\gamma_{1,k}|\cdot\frac{f_2-d_2\Delta f_2}{d_2}+|\gamma_{2,k}|\cdot(\Delta f_1+\Delta f_2) \tag{39}$$

$$=\sum_{j=1}^{2}|\gamma_{1,k}|(f_j'-\frac{(d_j-2)f_j}{d_j})+|\gamma_{2,k}|\cdot(f_j-f_j') \tag{40}$$

$$\overset{(d)}{\leq}\max(4|\gamma_{1,k}|,2(|\gamma_{1,k}|+|\gamma_{2,k}|))\cdot\eta\leq\max(4\gamma_{max}^{(1)},2\gamma_{max})\cdot\eta \tag{41}$$

where (a) follows from the non-negativity of the function $f$, and (b) is derived from $|A'\cap C|=d_1-1$ and $|B'\cap C|=d_2-1$. Equality is obtained when $C=\emptyset$, i.e., when there are no common neighbors between nodes 1 and 2. (c) is obtained from **Result 1** and (d) originates from **Result 2**.

**Result 1.** Consistent with the above notations, we have $f_j-d_j\Delta f_j\geq 0$ for $j\in\{1,2\}$.

**Results 2.** Consistent with the above notations, we have $|\gamma_{1,k}|(f_j'-\frac{(d_j-2)f_j}{d_j})+|\gamma_{2,k}|(f_j-f_j')\leq \max(2|\gamma_{1,k}|,|\gamma_{1,k}|+|\gamma_{2,k}|)\eta$ for $j\in\{1,2\}$.

**Proof of Result 1.** From the above definitions, we have $f_j=[f(\mathbf{x})]_j=\min(\max(x_j,0),\eta\cdot d_j)$ and $f_j-d_j\Delta f_j=d_jf_j'-(d_j-1)f_j$ where $x_j$ is the $j$-th entry of $\mathbf{x}$. Now, consider the following cases:

- When $x_j\leq 0$, we have $f_j-d_j\Delta f_j=0$.
- When $x_j\in(0,(d_j-1)\eta]$, it follows that $f_j-d_j\Delta f_j=d_j\eta x_j-(d_j-1)\eta x_j=\eta x_j\geq 0$.
- When $x_j\in((d_j-1)\eta,d_j\eta]$, we obtain $f_j-d_j\Delta f_j=(d_j-1)(d_j-x_j)\eta^2\geq 0$.
- When $x_j>d_j$, $f_j-d_j\Delta f_j=0$.

In conclusion, based on the above cases, the result is proven.

**Proof of Result 2.** We consider different cases:

- When $x_j \leq 0$, $|\gamma_{1,k}|(f'_j - \frac{(d_j-2)f_j}{d_j}) + |\gamma_{2,k}|(f_j - f'_j) = 0$.

- When $x_j \in (0, (d_j - 1)\eta]$, we have

$$|\gamma_{1,k}|(f'_j - \frac{(d_j - 2)f_j}{d_j}) + |\gamma_{2,k}|(f_j - f'_j) = \frac{2|\gamma_{1,k}|\eta[\mathbf{x}]_j}{d_j} \leq \frac{2|\gamma_{1,k}|(d_j - 1)\eta}{d_j} \tag{42}$$

- When $x_j \in ((d_j - 1)\eta, d_j\eta]$, we obtain

$$|\gamma_{1,k}|(f'_j - \frac{(d_j - 2)f_j}{d_j}) + |\gamma_{2,k}|(f_j - f'_j) \tag{43}$$

$$= |\gamma_{1,k}|\eta \left( \frac{(d_j - 1)d_j - (d_j - 2)x_j}{d_j} \right) + |\gamma_{2,k}|\eta(x_j - (d_j - 1)) \tag{44}$$

$$\leq \max(\frac{2|\gamma_{1,k}|(d_j - 1)\eta}{d_j}, (|\gamma_{1,k}| + |\gamma_{2,k}|)\eta) \leq \max(2|\gamma_{1,k}|, |\gamma_{1,k}| + |\gamma_{2,k}|)\eta \tag{45}$$

Note that, based on the derivations in $(b)$ and $(d)$, we conclude that the bound is asymptotically tight, with the worst-case scenario occurring when the nodes connected by the perturbed edge have no common neighbors, and their degrees increase to infinity.

Additionally, when $d_1 = 1$ or $d_2 = 1$, it can be directly shown that the distortion is upper bounded by $\max(\gamma_{\max}^{(1)}, 2\gamma_{\max}) \cdot \eta$. Note that if we further incorporate Euclidean projection onto $\ell_1$ ball, we can regard $\mathscr{P}_\mathcal{B}(\mathbf{x})$ as a single input without altering the overall bound.

### F.2 Proof of Lemma B.2

To prove this lemma, we introduce result on the Rényi divergence for high-dimensional Laplacian distributions with shift. The details of this result and its proof are presented following the lemma.

**Result:** Given a shift $\mathbf{h} \in \mathbb{R}^{|\mathcal{V}|}$, for two Laplacian distributions $\mathcal{L}(\mathbf{0}, \sigma)$ and $\mathcal{L}(\mathbf{h}, \sigma)$, if $\|\mathbf{h}\|_1 \leq \rho$,

$$\mathcal{D}_\alpha(\mathcal{L}(\mathbf{0}, \sigma)\|\mathcal{L}(\mathbf{h}, \sigma)) \leq \frac{1}{\alpha - 1} \ln(\frac{\alpha}{2\alpha - 1}\exp(\frac{\alpha - 1}{\sigma}\rho) + \frac{\alpha - 1}{2\alpha - 1}\exp(-\frac{\alpha}{\sigma}\rho)) \tag{46}$$

**Lemma Proof.** With the above result, we can upper bound the Rényi divergence over joint noise distributions. For $\tau \geq 0$,

$$\mathcal{D}_\alpha(\xi_{\tau+1:K}^{(2)}\|\tilde{\xi}_{\tau+1:K}^{\prime(2)}) \overset{(a)}{\leq} \sum_{k=\tau+1}^{K} \sup_{\zeta_{\tau+1:k-1}} \mathcal{D}_\alpha(\xi_k^{(2)}|\xi_{\tau+1:k-1}^{(2)} = \zeta_{\tau+1:k-1}\|\tilde{\xi}_k^{\prime(2)}|\tilde{\xi}_{\tau+1:k-1}^{\prime(2)} = \zeta_{\tau+1:k-1}) \tag{47}$$

$$\overset{(b)}{\leq} \sum_{k=\tau+1}^{K} \mathcal{D}_\alpha(\mathcal{L}(\mathbf{0}, \sigma_k)\|\mathcal{L}(\mathbf{h}_k, \sigma_k)) \overset{(c)}{\leq} \sum_{k=\tau+1}^{K} g_\alpha(\sigma_k, \rho_{\mathrm{diff}}) \tag{48}$$

where $(a)$ arises from strong composition rule for Rényi divergence (Lemma A.4), $(b)$ is from the definition of $\xi_k^{(2)}, \tilde{\xi}_k^{(2)}$ with shift $\mathbf{h}_k = (\phi_k \circ f)(\mathbf{x}) - (\phi'_k \circ f')(\mathbf{x})$, and $(c)$ is derived by combining the results that the $\ell_1$ norm of the shift is upper bounded by $\|\mathbf{h}_k\|_1 \leq \rho_{\mathrm{diff}} = \max(4\gamma_{\max}^{(1)}, 2\gamma_{\max})$ (Lemma B.1) and the above Laplace bound under Rényi divergence (Eq. (46)) with $g_\alpha(\sigma, \rho) = \frac{1}{\alpha-1} \ln(\frac{\alpha}{2\alpha-1}\exp(\frac{\alpha-1}{\sigma}\rho) + \frac{\alpha-1}{2\alpha-1}\exp(-\frac{\alpha}{\sigma}\rho_{\mathrm{diff}}))$.

Summarizing the above, we prove the lemma. Note that further considering a projection operator does not affect the distortion, as established in Lemma B.1, and thus leaves the bound derived here unchanged.

**Proof of Result.** Now consider Laplacian distribution, for $\mathbf{h} \in \mathbb{R}^{|\mathcal{V}|}$, let $h_i$ denote the $i$-th entry of the vector, we have

$$\mathcal{D}_\alpha(\mathcal{L}(\mathbf{0}, \sigma)\|\mathcal{L}(\mathbf{h}, \sigma)) \overset{(a)}{=} \sum_{i=1}^{|\mathcal{V}|} \frac{1}{\alpha - 1} \ln(\frac{\alpha}{2\alpha - 1}\exp(\frac{\alpha - 1}{\sigma}|h_i|) + \frac{\alpha - 1}{2\alpha - 1}\exp(-\frac{\alpha}{\sigma}|h_i|)) \tag{49}$$

where $(a)$ is from the additivity of Rényi divergence [60] and the Rényi divergence over one-dimensional Laplacian noise [61]. By considering the worst case $\mathbf{h}$, we can reformulate the problem as:

$$\underset{\mathbf{h}}{\text{maximize}} \sum_{i=1}^{|\mathcal{V}|} \frac{1}{\alpha - 1} \ln\left(\frac{\alpha}{2\alpha - 1} \exp\left(\frac{\alpha - 1}{\sigma}|h_i|\right) + \frac{\alpha - 1}{2\alpha - 1} \exp\left(-\frac{\alpha}{\sigma}|h_i|\right)\right) \tag{50}$$

$$\text{s.t. } \|\mathbf{h}\|_1 \leq \rho \tag{51}$$

For the above optimization problem, let $\lambda_1 = \frac{\alpha}{2\alpha-1}, \lambda_2 = \frac{\alpha-1}{\sigma}, \lambda_3 = \frac{\alpha-1}{2\alpha-1}$ and $\lambda_4 = \frac{\alpha}{\sigma}$. Define $f(h_i) = \frac{1}{\alpha-1} \ln(\lambda_1 \exp(\lambda_2 \cdot h_i) + \lambda_3 \exp(-\lambda_4 \cdot h_i))$. Consider the gradient term $\nabla \mathcal{D}_\alpha(\mathcal{L}(\mathbf{0}, \sigma) \| \mathcal{L}(\mathbf{h}, \sigma))$:

$$\frac{\partial \mathcal{D}_\alpha(\mathcal{L}(\mathbf{0}, \sigma) \| \mathcal{L}(\mathbf{h}, \sigma))}{\partial h_i} = \frac{\partial f(h_i)}{\partial h_i} = \frac{\lambda_1 \lambda_2 \exp(\lambda_2 h_i) - \lambda_3 \lambda_4 \exp(-\lambda_4 h_i)}{\lambda_1 \exp(\lambda_2 h_i) + \lambda_3 \exp(-\lambda_4 h_i)} \tag{52}$$

Further for Hessian matrix $\nabla^2 \mathcal{D}_\alpha(\mathcal{L}(\mathbf{0}, \sigma) \| \mathcal{L}(\mathbf{h}, \sigma))$,

$$\frac{\partial^2 \mathcal{D}_\alpha(\mathcal{L}(\mathbf{0}, \sigma) \| \mathcal{L}(\mathbf{h}, \sigma))}{\partial h_i^2} = \frac{\lambda_1 \lambda_3 (\lambda_2 + \lambda_4)^2 \exp(\lambda_2 h_i) \exp(-\lambda_4 h_i)}{\lambda_1 \exp(\lambda_2 h_i) + \lambda_3 \exp(-\lambda_4 h_i)} > 0, \tag{53}$$

$$\frac{\partial^2 \mathcal{D}_\alpha(\mathcal{L}(\mathbf{0}, \sigma) \| \mathcal{L}(\mathbf{h}, \sigma))}{\partial h_i h_j} = 0, i \neq j. \tag{54}$$

From the eigenvalue criterion for positive definiteness, $\nabla^2 \mathcal{D}_\alpha(\mathcal{L}(\mathbf{0}, \sigma) \| \mathcal{L}(\mathbf{h}, \sigma)) \succ 0$, i.e. $\mathcal{D}_\alpha(\mathcal{L}(\mathbf{0}, \sigma) \| \mathcal{L}(\mathbf{h}, \sigma))$ is a convex function. As the feasible set is convex, maximum is obtained on the boundary of feasible set. Thus, the problem can be further formulated as:

$$\underset{\mathbf{h}}{\text{maximize}} \sum_{i=1}^{|\mathcal{V}|} \frac{1}{\alpha - 1} \ln\left(\frac{\alpha}{2\alpha - 1} \exp\left(\frac{\alpha - 1}{\sigma}|h_i|\right) + \frac{\alpha - 1}{2\alpha - 1} \exp\left(-\frac{\alpha}{\sigma}|h_i|\right)\right) \text{ s.t. } \|\mathbf{h}\|_1 = \rho \tag{55}$$

Next, we use the adjustment method to solve the above objective methods. First, we define $L(h_1, h_2, \ldots, h_{|\mathcal{V}|}) = \sum_{i=1}^{|\mathcal{V}|} f(h_i)$ and we fix $h_3, \ldots, h_{|\mathcal{V}|}$. We aim to optimize:

$$\underset{h_1, h_2}{\text{maximize}} \, L(h_1, h_2, h_3, \ldots, h_{|\mathcal{V}|}) \tag{56}$$

with respect to $h_1$ and $h_2$, considering the constraints $\|\mathbf{h}\|_1 = \rho$. This is equivalent to

$$\underset{h_1, h_2}{\text{maximize}} \sum_{i=1}^{2} L(h_1, h_2, \ldots, h_{|\mathcal{V}|}), \text{ s.t. } h_1 + h_2 = \rho - \sum_{i=3}^{|\mathcal{V}|} h_i. \tag{57}$$

Define $c_t = \rho - \sum_{i=t}^{\mathcal{V}} h_i$. Since $c_3$ is fixed, and $h_2 = c_3 - h_1$, the objective function become a univariate function, by calculating the derivative

$$\frac{\partial L}{\partial h_1} = \frac{(\lambda_1 \lambda_3 \lambda_4 + \lambda_1 \lambda_2 \lambda_3)(\exp(\lambda_2 h_1 + \lambda_4(h_1 - c_3)) - \exp(\lambda_2(c_3 - h_1) - \lambda_4 h_1))}{(\lambda_1 \exp(\lambda_2 h_1 + \lambda_3 \exp(-\lambda_4 h_1)))(\lambda_1 \exp(\lambda_2(c_3 - h_1)) + \lambda_3 \exp(-\lambda_4(c_3 - h_1)))} \tag{58}$$

The derivative $\frac{\partial L}{\partial h_1}$ reaches 0 when $h_1 = \frac{c_3}{2}$. Furthermore, $\frac{\partial L}{\partial h_1} < 0$ for $h_1 < \frac{c_3}{2}$, and $\frac{\partial L}{\partial h_1} > 0$ for $h_1 > \frac{c_3}{2}$. Thus, the maximum value of the function is attained at the endpoints.

$$\underset{h_1, h_2}{\text{maximize}} \, L(h_1, h_2, h_3, \ldots, h_{|\mathcal{V}|}) \leq \max\left[L(c_3, 0, h_3, \ldots, h_{|\mathcal{V}|}), L(0, c_3, h_3, \ldots, h_{|\mathcal{V}|})\right] \tag{59}$$

As the function is symmetric, two endpoints attain the same value, i.e. $L(c_3, 0, h_3, \ldots, h_{|\mathcal{V}|}) = L(0, c_3, h_3, \ldots, h_{|\mathcal{V}|})$. Now, we aim to maximize the objective function by each time adjustment two variables and fixed the rest variables unchanged:

$$\underset{h_1, h_2, \ldots, h_{|\mathcal{V}|}}{\text{maximize}} \, L(h_1, h_2, h_3, \ldots, h_{|\mathcal{V}|}) = \underset{\substack{h_1, h_2, h_3, \ldots, h_{|\mathcal{V}|} \\ h_1 + h_2 + c_3 = \rho}}{\text{maximize}} \, L(h_1, h_2, h_3, \ldots, h_{|\mathcal{V}|}) \tag{60}$$

$$\leq \underset{h_3, \ldots, h_{|\mathcal{V}|}}{\text{maximize}} \, \underset{\substack{h_1, h_2 \\ h_1 + h_2 = \rho - c_3}}{\text{maximize}} \, L(h_1, h_2, h_3, \ldots, h_{|\mathcal{V}|}) \tag{61}$$

$$\overset{(a)}{\leq} \underset{h_3, h_4, \ldots, h_{|\mathcal{V}|}}{\text{maximize}} \; L(\rho - c_3, 0, h_3, \ldots, h_{|\mathcal{V}|}) \leq \underset{h_4, \ldots, h_{|\mathcal{V}|}}{\text{maximize}} \; L(\rho - c_4, 0, 0, \ldots, h_{|\mathcal{V}|}) \tag{62}$$

$$\leq \cdots \leq L(\rho, 0, 0, \ldots, 0) = \frac{1}{\alpha - 1} \ln\left(\frac{\alpha}{2\alpha - 1} \exp\left(\frac{\alpha - 1}{\sigma}\rho\right) + \frac{\alpha - 1}{2\alpha - 1} \exp\left(-\frac{\alpha}{\sigma}\right)\rho\right) \tag{63}$$

where $(a)$ is from Eq. (59), and for each inequality, we adjust the values of $h_1$ and $h_i$ to maximized the objective function while keep the rest variables fixed. From the above reasoning, we maximized the Rényi divergence over two Laplacians with shift.

### F.3 Proof of Lemma B.3.

To prove the upper bound of $\sup_{\zeta_{\tau+1:K}} \mathcal{D}_\alpha(\mathbf{s}_K | \xi^{(2)}_{\tau+1:K} = \zeta_{\tau+1:K} \| \mathbf{s}'_K | \tilde{\xi}'^{(2)}_{\tau+1:K} = \zeta_{\tau+1:K})$, we mainly leverage the definition of shifted Rényi divergence (Def. A.1) with two properties under noise convolution and contraction mapping (Lemma A.1 and Lemma A.2). Specifically, recall that $\mathcal{R}_\alpha(\sigma, \rho) = \sup_{\mathbf{r}: \|\mathbf{r}\| \leq \rho} \mathcal{D}_\alpha(\xi + \mathbf{r} \| \xi)$ where $\xi \sim \mathcal{L}(\mathbf{0}, \sigma)$, define $\varphi_k = \phi_k \circ f$, for any $\zeta_{\tau+1:K}$, define $w_K = 0$,

$$\mathcal{D}_\alpha(\mathbf{s}_K | \xi^{(2)}_{\tau+1:K} = \zeta_{\tau+1:K} \| \mathbf{s}'_K | \tilde{\xi}'^{(2)}_{\tau+1:K} = \zeta_{\tau+1:K}) \tag{64}$$

$$= \mathcal{D}^{(w_K)}_\alpha(\mathbf{s}_K | \xi^{(2)}_{\tau+1:K} = \zeta_{\tau+1:K} \| \mathbf{s}'_K | \tilde{\xi}'^{(2)}_{\tau+1:K} = \zeta_{\tau+1:K}) \tag{65}$$

$$\overset{(a)}{\leq} \mathcal{D}^{(w_K + \mathbf{a}_K)}_\alpha(\varphi_K(\mathbf{s}_{K-1}) | \xi^{(2)}_{\tau+1:K-1} = \zeta_{\tau+1:K-1} \| \varphi_K(\mathbf{s}'_K) | \tilde{\xi}'^{(2)}_{\tau+1:K-1} = \zeta_{\tau+1:K-1}) + \mathcal{R}_\alpha(\sigma_K, \mathbf{a}_K) \tag{66}$$

$$\overset{(b)}{\leq} \mathcal{D}^{\left(\frac{w_K + \mathbf{a}_K}{\gamma_{\max}}\right)}_\alpha(\mathbf{s}_{K-1} | \xi^{(2)}_{\tau+1:K-1} = \zeta_{\tau+1:K-1} \| \mathbf{s}'_K | \tilde{\xi}'^{(2)}_{\tau+1:K-1} = \zeta_{\tau+1:K-1}) + \mathcal{R}_\alpha(\sigma_K, \mathbf{a}_K) \tag{67}$$

$$\overset{w_{K-1} = \frac{w_K + \mathbf{a}_K}{\gamma_{\max}}}{=\!=\!=\!=\!=\!=\!=} \mathcal{D}^{(w_{K-1})}_\alpha(\mathbf{s}_{K-1} | \xi^{(2)}_{\tau+1:K-1} = \zeta_{\tau+1:K-1} \| \mathbf{s}'_K | \tilde{\xi}'^{(2)}_{\tau+1:K-1} = \zeta_{\tau+1:K-1}) + \mathcal{R}_\alpha(\sigma_K, \mathbf{a}_K) \tag{68}$$

$$\leq \cdots \tag{69}$$

$$\leq \mathcal{D}^{(w_{\tau+1})}_\alpha(\mathbf{s}_{\tau+1} | \xi^{(2)}_{\tau+1} = \zeta_{\tau+1} \| \mathbf{s}'_{\tau+1} | \tilde{\xi}'^{(2)}_{\tau+1} = \zeta_{\tau+1}) + \sum_{k=\tau+2}^{K} \mathcal{R}_\alpha(\sigma_k, \mathbf{a}_k) \tag{70}$$

$$\leq \mathcal{D}^{(w_\tau)}_\alpha(\mathbf{s}_\tau \| \mathbf{s}'_\tau) + \sum_{k=\tau+1}^{K} \mathcal{R}_\alpha(\sigma_k, \mathbf{a}_k) \tag{71}$$

where $(a)$ is from Lemma A.1, and $(b)$ is derived from Lemma A.2 as $\varphi_k$ is $\gamma_{\max}$-contractive (composition of two contractions). Further, we have

$$\begin{cases} w_{\tau+1} = \gamma_{\max} w_\tau - \mathbf{a}_{\tau+1} \\ w_{\tau+2} = \gamma_{\max} w_{\tau+1} - \mathbf{a}_{\tau+2} = \gamma_{\max}^2 w_\tau - \gamma_{\max} \mathbf{a}_{\tau+1} - \mathbf{a}_{\tau+2} \\ \cdots = \cdots \\ w_K = \gamma_{\max}^{K-\tau} w_\tau - \sum_{k=0}^{K-\tau-1} \gamma_{\max}^k \mathbf{a}_{K-k} \end{cases} \tag{72}$$

By setting $\mathbf{a}_{\tau+1} = 0, \ldots, \mathbf{a}_{K-1} = 0$, we have $\mathbf{a}_K = \gamma_{\max}^{K-\tau} w_\tau$. Therefore, we have

$$\mathcal{D}_\alpha(\mathbf{s}_K | \xi^{(2)}_{\tau+1:K} = \zeta_{\tau+1:K} \| \mathbf{s}'_K | \tilde{\xi}'^{(2)}_{\tau+1:K} = \zeta_{\tau+1:K}) \leq \mathcal{D}^{(w_\tau)}_\alpha(\mathbf{s}_\tau \| \mathbf{s}'_\tau) + \mathcal{R}_\alpha(\sigma_K, \gamma_{\max}^{K-\tau} w_\tau) \tag{73}$$

From the result in the proof of Lemma B.2, $\mathcal{R}_\alpha(\sigma_K, \gamma_{\max}^{K-\tau} w_\tau) = g_\alpha(\sigma_K, \gamma_{\max}^{K-\tau} w_\tau)$. Summarizing the above, we obtain

$$\mathcal{D}_\alpha(\mathbf{s}_K | \xi^{(2)}_{\tau+1:K} = \zeta_{\tau+1:K} \| \mathbf{s}'_K | \tilde{\xi}'^{(2)}_{\tau+1:K} = \zeta_{\tau+1:K}) \leq \mathcal{D}^{(w_\tau)}_\alpha(\mathbf{s}_\tau \| \mathbf{s}'_\tau) + g_\alpha(\sigma_K, \gamma_{\max}^{K-\tau} w_\tau) \tag{74}$$

Further, we point out that incorporating the projection operator, i.e., $\varphi_k = \phi_k \circ f \circ \mathscr{P}_\mathcal{B}$, does not affect the contractiveness of the mapping. Indeed, projecting a vector onto the $\ell_1$-ball via Euclidean projection yields a nonexpansive operator under the $\ell_1$ norm [62], and thus this extension does not alter the bound.

## F.4  Proof of Lemma B.4.

To prove this lemma, we consider tracking the $\infty$-Wasserstein distance of the coupled iterates. Given $\mathscr{D}_{K,\Pi}$ and $\mathscr{D}'_{K,\Pi}$, for $\tau \geq 1$, recall that for any $k(k \geq 1)$,

$$\mathscr{D}_{K,\Pi}(\mathbf{s}) : \mathbf{s}_k = \phi_k(f(\mathbf{s}_{k-1})) + \xi_k^{(1)} + \xi_k^{(2)}, \quad \mathscr{D}'_{K,\Pi}(\mathbf{s}) : \mathbf{s}'_k = \phi'_k(f'(\mathbf{s}'_{k-1})) + \xi_k'^{(1)} + \tilde{\xi}_k'^{(2)}. \quad (75)$$

For any step $k(k \geq 1)$, let $\mu_k$ and $\nu_k$ denote the distribution of $\mathbf{s}_k$ and $\mathbf{s}'_k$ respectively. Further, define $\tilde{\mu}_k$ and $\tilde{\nu}_k$ denote the distribution of $\mathbf{S}_k = (\xi_{1:k}, \tilde{\xi}_{1:k})$ and $\mathbf{S}'_k = (\xi'_{1:k}, \tilde{\xi}'_{1:k})$, respectively. For step 1, we have

$$W_\infty(\mu_1, \nu_1) = \inf_{\pi_1 \in \Gamma(\mu_1, \nu_1)} \operatorname*{ess\,sup}_{(\mathbf{s}_1, \mathbf{s}'_1) \sim \pi_1} \|\mathbf{s}_1 - \mathbf{s}'_1\|_1 \quad (76)$$

$$= \inf_{\tilde{\pi}_1 \in \Gamma(\tilde{\mu}_1, \tilde{\nu}_1)} \operatorname*{ess\,sup}_{(\mathbf{S}_1, \mathbf{S}'_1) \sim \tilde{\pi}_1} \|\phi_1(f(\mathbf{s}_0)) + \xi_1^{(1)} + \xi_1^{(2)} - \phi'_1(f'(\mathbf{s}_0)) - \xi_1'^{(1)} - \xi_1'^{(2)}\|_1 \quad (77)$$

$$\overset{(a)}{\leq} \operatorname*{ess\,sup}_{(\mathbf{S}_1, \mathbf{S}'_1) \sim \tilde{\pi}_1^*} \|\phi_1(f(\mathbf{s}_0)) - \phi'_1(f'(\mathbf{s}_0))\|_1 \leq \rho_{\text{diff}} \quad (78)$$

where $(a)$ is from selecting a coupling $\tilde{\pi}^*$ such that r.v.s $\mathbf{S}_1$ and $\mathbf{S}'_1$ are identical.

Next, for any $\tau$, following the above procedure, define $\tilde{\pi}_k^* \in \Gamma(\mu_k, \nu_k)$ such that $\mathbf{S}_k, \mathbf{S}'_k$ are identical, we have

$$W_\infty(\mu_\tau, \nu_\tau) = \inf_{\pi_\tau \in \Gamma(\mu_\tau, \nu_\tau)} \operatorname*{ess\,sup}_{(\mathbf{s}_\tau, \mathbf{s}'_\tau) \sim \pi_\tau} \|\mathbf{s}_\tau - \mathbf{s}'_\tau\|_1 \quad (79)$$

$$\leq \inf_{\tilde{\pi}_\tau \in \Gamma(\tilde{\mu}_\tau, \tilde{\nu}_\tau)} \operatorname*{ess\,sup}_{(\mathbf{S}_\tau, \mathbf{S}'_\tau) \sim \tilde{\pi}_\tau} \|\phi_\tau(f_\tau(\mathbf{s}_{\tau-1})) + \xi_\tau^{(1)} + \xi_\tau^{(2)} - \phi'_\tau(f'_\tau(\mathbf{s}'_{\tau-1})) - \xi_\tau'^{(1)} - \xi_\tau'^{(2)}\|_1 \quad (80)$$

$$\leq \operatorname*{ess\,sup}_{(\mathbf{S}_\tau, \mathbf{S}'_\tau) \sim \tilde{\pi}_\tau^*} \|\phi_\tau(f_\tau(\mathbf{s}_{\tau-1})) + \xi_\tau^{(1)} + \xi_\tau^{(2)} - \phi'_\tau(f'_\tau(\mathbf{s}'_{\tau-1})) - \xi_\tau'^{(1)} - \xi_\tau'^{(2)}\|_1 \quad (81)$$

$$= \operatorname*{ess\,sup}_{(\mathbf{S}_{\tau-1}, \mathbf{S}'_{\tau-1}) \sim \tilde{\pi}_{\tau-1}^*} \|\phi_\tau(f_\tau(\mathbf{s}_{\tau-1})) - \phi'_\tau(f'_\tau(\mathbf{s}'_{\tau-1}))\|_1 \quad (82)$$

Following the analysis in Lemma B.1, by induction,

$$\operatorname*{ess\,sup}_{(\mathbf{S}_{\tau-1}, \mathbf{S}'_{\tau-1}) \sim \tilde{\pi}_{\tau-1}^*} \|\phi_\tau(f_\tau(\mathbf{s}_{\tau-1})) - \phi'_\tau(f'_\tau(\mathbf{s}'_{\tau-1}))\|_1 \quad (83)$$

$$\leq \operatorname*{ess\,sup}_{(\mathbf{S}_{\tau-1}, \mathbf{S}'_{\tau-1}) \sim \tilde{\pi}_{\tau-1}^*} \|\phi_\tau(f_\tau(\mathbf{s}_{\tau-1})) - \phi_\tau(f_\tau(\mathbf{s}'_{\tau-1}))\|_1 + \|\phi_\tau(f_\tau(\mathbf{s}'_{\tau-1})) - \phi'_\tau(f'_\tau(\mathbf{s}'_{\tau-1}))\|_1 \quad (84)$$

$$\overset{(a)}{\leq} \operatorname*{ess\,sup}_{(\mathbf{S}_{\tau-1}, \mathbf{S}'_{\tau-1}) \sim \tilde{\pi}_{\tau-1}^*} \gamma_{\max} \cdot \|\mathbf{s}_{\tau-1} - \mathbf{s}'_{\tau-1}\|_1 + \rho_{\text{diff}} \quad (85)$$

$$\leq \cdots \quad (86)$$

$$\overset{(b)}{\leq} \operatorname*{ess\,sup}_{(\mathbf{S}_1, \mathbf{S}'_1) \sim \tilde{\pi}_1^*} \gamma_{\max}^{\tau-1} \cdot \|\mathbf{s}_1 - \mathbf{s}'_1\|_1 + \rho_{\text{diff}} \left(1 + \gamma_{\max} + \cdots + \gamma_{\max}^{\tau-2}\right) \quad (87)$$

$$\overset{(c)}{\leq} \gamma_{\max}^{\tau-1} \cdot \frac{(1 - \gamma_{\max}) \cdot \rho_{\text{diff}}}{1 - \gamma_{\max}} + \frac{\rho_{\text{diff}} \cdot (1 - \gamma_{\max}^{\tau-1})}{1 - \gamma_{\max}} \quad (88)$$

$$= \frac{\rho_{\text{diff}} \cdot (1 - \gamma_{\max}^\tau)}{1 - \gamma_{\max}} \quad (89)$$

where $(a)$ is from Lemma B.1, $(b)$ is from induction, and $(c)$ arises from Eq. (78).

Therefore,

$$W_\infty(\mu_\tau, \nu_\tau) \leq \frac{\rho_{\text{diff}} \cdot (1 - \gamma_{\max}^\tau)}{1 - \gamma_{\max}} \quad (90)$$

Further, when we set $w_\tau = \frac{\rho_{\text{diff}} \cdot (1 - \gamma_{\max}^\tau)}{1 - \gamma_{\max}}$, we have

$$\mathcal{D}_\alpha^{(w_\tau)}(\mathbf{s}_\tau \| \mathbf{s}'_\tau) = \inf_{\mu'_\tau : W_\infty(\mu_\tau, \mu'_\tau) \leq w_\tau} \mathcal{D}_\alpha(\mu'_\tau \| \nu_\tau) = 0 \quad (91)$$

where the equality is achieved by selecting $\mu'_\tau = \nu_\tau$.

