# OpenReview forum: "Differentially Private Graph Diffusion with Applications in Personalized PageRanks"
_NeurIPS.cc/2024/Conference — NeurIPS 2024 poster_

### Official Review · Reviewer_y6QA · 2024-07-02

**Soundness:** 4
**Presentation:** 4
**Contribution:** 3
**Rating:** 8
**Confidence:** 4

**Summary:**

The authors study the important problem graph learning methods (graph diffusion) with differential privacy. There is limited work in this important problem space (one exception being some work on PPR with DP). The authors present a non-trivial use of the PABI framework which uses Contractive Noisy Iterations to prove differential privacy for a process that adds noise iteratively and applies a contractive function. This is a novel and interesting result that might have impact on a variety of learning problems beyond DP.
The authors prove formally privacy guarantees of the method. Then the perform an empirical analysis on real data of moderate size and show that their method exceeds all existing baselines (prior DP PPR paper and trivial edge flipping baseline).

This paper addresses an important problem with novel and practical ideas showing improvements both theoretically and practically.

**Strengths:**

Important problem: graph learning with privacy
Novel algorithmic method that appears to be general enough to have impact on multiple graph problems.
Theoretical guarantees for privacy
Good evidence of improving empirically over all prior baselines

**Weaknesses:**

The paper lacks theoretical lower bounds showing the method is tight.

**Questions:**

No question

---

> ### Author Rebuttal · Authors · 2024-08-05
>
> We extend our gratitude to Reviewer y6QA for appreciating our theoretical and practical contributions and supporting the acceptance of this paper. Here, we are to respond to the weaknesses proposed by Reviewer y6QA.
>
> >The paper lacks theoretical lower bounds showing the method is tight. (W1)
>
> The Reviewer y6QA raises an insightful point. We agree that establishing a privacy lower bound is essential for evaluating the tightness of our framework and identifying areas for improvement. The original PABI analysis on noisy SGD [1] provides an order-tight lower bound by constructing a dataset with zero loss and an adjacent dataset with a bias introduced by a differing data sample. The parameter update processes in these adjacent datasets are thus characterized by a symmetric random process and a biased random process. The distinction between these processes is determined by test positivity [1], as one is symmetric and the other is biased. We believe a similar approach could be applied to our graph diffusion analysis, and we plan to explore this in future work.
>
> [1] Altschuler et al. On the Privacy of Noisy Stochastic Gradient Descent for Convex Optimization.

---

> > ### Comment · Reviewer_y6QA · 2024-08-07
> >
> > Thanks for the rebuttal.

---

### Official Review · Reviewer_dwyJ · 2024-07-08

**Soundness:** 3
**Presentation:** 3
**Contribution:** 3
**Rating:** 5
**Confidence:** 3

**Summary:**

The paper titled "Differentially Private Graph Diffusion with Applications in Personalized PageRanks" proposes a novel graph diffusion framework that ensures edge-level differential privacy (DP) by injecting Laplace noise into the diffusion process. This framework leverages Privacy Amplification by Iteration (PABI) and introduces a new method for tracking privacy leakage using the ∞-Wasserstein distance. The proposed approach is evaluated in the context of Personalized PageRank (PPR) computation and demonstrates superior performance under stringent privacy conditions compared to existing methods.

**Strengths:**

The paper introduces a novel method for achieving edge-level differential privacy in graph diffusion processes by incorporating Laplace noise and utilizing Privacy Amplification by Iteration (PABI). This approach is innovative and adds to the existing body of knowledge on privacy-preserving graph algorithms.

The authors provide a thorough theoretical analysis of the privacy guarantees of their method, including a novel ∞-Wasserstein distance tracking method to tighten privacy leakage bounds. This rigorous analysis enhances the credibility and potential impact of the work.

The application of the proposed framework to Personalized PageRank (PPR) is highly relevant, as PPR is widely used in various real-world applications such as recommendation systems, community detection, and targeted marketing.

 The paper includes extensive experiments on real-world datasets (BlogCatalog, Flickr, TheMarker) demonstrating that the proposed method achieves better privacy-utility trade-offs compared to baseline methods, particularly under stringent privacy settings.

**Weaknesses:**

Complexity of Implementation: The proposed method involves complex theoretical constructs such as PABI and ∞-Wasserstein distance tracking, which may pose challenges for implementation and adoption in practical applications.

Specific Focus on PPR: While the application to Personalized PageRank is well-justified, the paper's focus on a single type of graph diffusion might limit its generalizability.

Scalability Concerns: The scalability of the proposed method, particularly for very large graphs, is not thoroughly addressed. Additional experiments or discussions on the computational efficiency and scalability of the approach would strengthen the paper.

**Questions:**

Generalizability: Can the proposed framework be easily adapted to other types of graph diffusion processes beyond Personalized PageRank? If so, what modifications would be necessary?

Scalability: How does the proposed method scale with increasing graph size and complexity? Are there any optimizations or approximations that could improve scalability?

Parameter Tuning: How sensitive is the performance of the proposed method to the choice of the noise scale σ and the degree-based thresholding parameter? Are there guidelines for selecting these parameters in practice?

**Limitations:**

See above

---

> ### Author Rebuttal · Authors · 2024-08-05
>
> We greatly thank Reviewer dwyJ for appreciating the novelty of our method and the corresponding theoretical analysis, and acknowledging the practical importance of the problem we studied. Here, we are to respond to the questions and weaknesses proposed by Reviewer dwyJ.
>
> >Complexity of Implementation. (W1)
>
> We appreciate the reviewer’s question. Our framework comprises two main components: noisy graph diffusion algorithm and noise calibration. The noisy graph diffusion is straightforward and efficient to implement, involving only thresholding and noise injection after each diffusion step. The noise calibration is derived from our theoretical results (Theorem 1 & 4), which is computed before our graph diffusion algorithm. The efficiency of this part is further discussed in the text following Equation 6. For a given noisy graph diffusion algorithm (with a predefined thresholding function), our noise calibration method only requires the privacy budget to determine the appropriate noise scale, which the algorithm then outputs directly. To facilitate practical use, we will include the pseudocode for both components in the appendix. Additionally, we have released the code for noise calibration, enabling practitioners to apply it directly for effective noise calibration.
>
> >Generalizability to other graph diffusions? (W2 & Q1)
>
> We thank Reviewer dwyJ for the insightful question. Yes, our framework (Theorem 1) includes broader graph diffusion processes, such as Global PageRanks [1] and Generalized PageRanks [2], with Heat Kernel diffusions [3] as a special case. Additionally, when viewing graph diffusion as a graph operator, our theoretical results can also be applied to graph convolution operators with non-linear transformations, provided these transformations satisfy certain Lipschitz continuity properties and do not require learnable parameters that necessitate backpropagation through the graph diffusion [4]. This approach further paves the way for obtaining edge-level private node embeddings for graph learning scenarios.
>
> [1] Brin et al. The anatomy of a large-scale hypertextual web search engine.
>
> [2] Li et al. Optimizing generalized pagerank methods for seed-expansion community detection.
>
> [3] Chung Fan. The heat kernel as the pagerank of a graph.
>
> [4] Sajadmanesh et al. GAP: Differentially Private Graph Neural Networks with Aggregation Perturbation.
>
> >The scalability of the proposed method. (Q2)
>
> We appreciate Reviewer dwyJ for raising this important question. In our paper, the largest graph dataset we consider is Flickr, with around 80k nodes and 6 million edges. For comparison, the largest dataset in previous research, Blogcatalog, contains around 10k nodes and 334k edges. We discuss the runtime in Appendix D.2 and Figure 7, where our method demonstrates comparable performance to the most efficient DP-PUSHFLOWCAP [1] method, completing a single trial on the Flickr dataset in under a minute. Specifically, on datasets with nodes ranging from 10k to 80k and edges from 334k to 6 million, our algorithm's runtime increases from 24s to 59s.
>
> Our algorithm consists of two main parts: noise calibration and noisy graph diffusion. Given a specific privacy budget and the thresholding parameter $\eta$, we first calibrate the noise scale according to Theorem 1 using bisection search (pseudocode will be included in the appendix in the revised version). We then perform graph diffusion followed by a fixed thresholding and noise injection at each step with a complexity of $O$(\# nodes), which is usually much smaller than $O$(\# edges). However, we acknowledge that exploring further techniques to enhance scalability could be an interesting direction for future work.
>
> [1] Epasto et al. Differentially private graph learning via sensitivity-bounded personalized pagerank.
>
> >Sensitivity of parameters. (Q3)
>
> We thank Reviewer dwyJ for highlighting this point. In our framework, the degree-based thresholding parameter $\eta$ is the only parameter that can be tuned, while the noise scale is determined by the privacy budget (illustrated in Theorem 1). In practice, once the privacy budget $\epsilon$ is set, the noise scale is determined by our theorems for the noisy graph diffusion process. Regarding the degree-based thresholding parameter $\eta$, we have illustrated its performance sensitivity in Appendix D.3, Figure 8. The results show that our framework is robust to the choice of $\eta$, with performance remaining stable when $\eta$ is selected within the range of $10^{-10}$ to $10^{-6}$, consistent with the recommendations in [1]. For practical purposes, we suggest selecting $\eta$ within this interval.
>
> [1] Epasto et al. Differentially private graph learning via sensitivity-bounded personalized pagerank.

---

### Official Review · Reviewer_sGeR · 2024-07-12

**Soundness:** 4
**Presentation:** 2
**Contribution:** 3
**Rating:** 5
**Confidence:** 2

**Summary:**

The paper presents a new differentially private personalized PageRank (PPR) algorithm. The paper extends and theoretically analyzes the privacy ampliﬁcation by iterations (PABI) technique applied to a graph diffusion setting. Rigorous theoretical results demonstrate the privacy guarantees achieved by the approach. An experimental evaluation shows that the proposed approach achieves better results than known algorithms

**Strengths:**

- The considered problem is important as diffusion vectors might be used to reveal the edges in the graph.
- The proposed technique appears to be interesting.
- The theoretical results and proof techniques are non-trivial and require in-depth subject knowledge.

**Weaknesses:**

The paper is extremely difficult to read. I admit I am no expert in the area but, for example, I didn't have a problem following Epasto et al. [20]. The paper is overloaded with advanced mathematical terms and concepts that are never properly defined. Maybe terms like Banach space, the diameter of a convex bounded set, contraction maps, Lipschitz continuous graph diffusions, essential supremum, etc are standard terms for a mathematical paper but, after all, NeurIPS is not a mathematical venue.  Even the $\infty$-Wasserstein distance is only defined in the appendix.
The paper would become much more accessible if the authors provided some intuition above the approach. For example, a paragraph can be devoted to Figure 1, a vanilla version of the main result simplifying some of the terms can be presented, pseudocode of the approach can be added to the appendix.

**Questions:**

- When computing Recall@R and NDCG@R, do you use as a ground truth the rankings computed by the non-private PPR algorithm, like in [20]?
- Would it be possible to show some theoretical results on how the quality of the diffusion vectors degrades by enhancing the privacy parameters, for some intuitive notion of "quality"?

**Limitations:**

I am not really happy with the discussion of limitations. The required noise to achieve privacy guarantees, as defined in (3), appears to be substantial. Also, as evident from Figure 5, the recall scores are considerably worse compared to the baselines. I want to emphasize that this is not meant as a criticism of the proposed algorithm, it clearly improves upon previous work. But the price that one has to pay for obtaining diffusion vectors with privacy guarantees is a limitation that needs to be acknowledged.

---

> ### Author Rebuttal · Authors · 2024-08-05
>
> We greatly thank Reviewer sGeR for appreciating our contributions to the algorithm and corresponding theoretical results. Here, we are to respond to the questions and weaknesses proposed by Reviewer sGeR.
>
> >The paper is difficult to read; consider simplifying terms, and including pseudocode. (W1)
>
> We sincerely thank Reviewer sGeR for the valuable feedback. To improve the accessibility of the paper, we will make the following adjustments in our revised version: (1) We remove some abstract notations, such as Banach space, and replaced them with specific terms, like the Euclidean space $R^{|\mathcal{V}|}$. (2) We plan to include definitions, such as Lipschitz continuity, the $\infty$-Wasserstein distance, in the main text. We will also expand the discussion around Figure 1 and provide additional explanations for our main results in Equation 5, if space allows. (3) We include the pseudocode for our private graph diffusion algorithm along with the corresponding privacy accounting method in the appendix.
>
> >Is non-private PPR used as ground truth for Recall@R and NDCG@R, like in [20]? (Q1)
>
> Yes, in our experiments, we use the rankings computed by the non-private PPR algorithm as the ground truth, consistent with the baseline approach described in [20].
>
>
> >Possible to measure utility degradation with increased privacy? (Q2)
>
> We appreciate Reviewer sGeR's insightful question. In the privacy literature, algorithms are typically developed to meet specific privacy requirements and are often validated empirically, as seen in works like [1]. Specifically, while addressing worst-case scenarios under privacy constraints is crucial, necessitating theoretical proofs, the utility of these methods is generally assessed through empirical evaluation. That being said, we acknowledge that in some cases, worst-case utility can also be significant. While our current work emphasizes empirical evaluation, we recognize the importance of exploring the theoretical relationship between privacy parameters and the quality degradation of noisy diffusion vectors—such as in terms of approximation error [2,3]. This exploration is planned as a direction for our future research, which could enhance understanding of how increased privacy impacts the quality of diffusion vectors and ensure robustness in more challenging scenarios.
>
> [1] Abadi et al. Deep learning with differential privacy.
>
> [2] Andersen et al. Using pagerank to locally partition a graph.
>
> [3] Hou et al. Massively parallel algorithms for personalized pagerank.
>
> >Discussion of limitations.(L1)
>
> Thanks Reviewer sGeR for pointing this out. We fully acknowledge that privacy is not without cost, and this phenomenon is commonly referred to as privacy-utility trade-offs. Our experiments (Fig. 4 & 5) demonstrate these trade-offs, where enhancing privacy budget $\epsilon$ results in some degradation of utility. Following your suggestion, we will further emphasize this point in the "Limitations" section of the paper.

---

### Official Review · Reviewer_ApdU · 2024-07-24

**Soundness:** 3
**Presentation:** 3
**Contribution:** 1
**Rating:** 4
**Confidence:** 3

**Summary:**

Graph diffusion iteratively propagates signals through the graph, that are subsequently used for real-world applications like personalized page rank. This paper develops edge-level Differentially Private (DP) guarantees for personalized PageRank using Laplace noise addition. Unlike traditional perturbation-based DP algorithms, the Laplace noise addition is done during the diffusion phase and not added directly to the output. This allows the algorithm to achieve better utility-privacy tradeoffs compared to existing approaches.

The authors identify the graph diffusion process as a composition of contraction maps and aim to extend the Privacy Amplification by Iteration (PABI), guarantees to the graph setting. To this end, the authors introduce degree-based thresholding of signals to bound distance between coupled diffusions over adjacent graphs. The authors prove the DP guarantees for the general setting of delayed noise injection where, noise is injected into the signals starting from diffusion iteration step “m”. They first bound the L1 norm between the two signals for all < m-th iterations. The Renyi Divergence (RD) between the final signal iterates are then upper bounded by the sum of shift absorption term and a PABI term using the properties of RD and the previous bound. The shift absorption term is finally bounded by the strong composition rule of RD and bounding the shift “h_k” per iteration. The PABI term is bounded by the infinite Wasserstein distance between the coupled iterates using the identity mapping.

The authors empirically demonstrate the utility gains (NDCG, RECALL) of the algorithm on BlogCatalog, Flickr and TheMarker datasets with respect to two baselines: DP-PUSHFLOWCAP and Edge-Flipping.

**Strengths:**

1. The algorithm performs Laplace noise injection during the signal diffusion phase, as opposed to output perturbation, resulting in better privacy utility tradeoffs.

2. Develops theoretical guarantees in epsilon-RDP based on tractable hyperparameters.

3. The method guarantees the RD Privacy budget remains bounded w.r.t the number of iterations.

**Weaknesses:**

1. Edge differential privacy is quite weak. I think the authors should try node differential privacy or atleast some stronger notion of privacy. Just with Edge DP, the privacy guarantee is quite weak.

2. The iterative privacy framework is quite well known since Thakurta et al.'s work. PageRank is well known iterative algorithm. In that sense, the contribution of the paper is limited. I was expecting to see some graph specific insights from either experiments or theory, which I did not find.

3. Since the algorithm is quite generic, can one extend it to learning or inference of a graph neural network?

The key weaknesses therefore are lack of novelty and impact.

**Questions:**

1. Since Theorem B.1. holds for any “m”, have you observed the utility privacy tradeoffs for different “m” values? Specifically are there any advantages to performing delayed noise injection instead of starting at m=0?

2. Graph prediction tasks are usually performed over node embeddings. These embeddings are typically generated by graph convolution operators which perform an operation similar to graph diffusion followed by a non-linearity. Is it possible to extend the DP guarantees to node embeddings in such settings? Such a guarantee can possibly extend the results in this paper to more general graph prediction tasks (other than PBR).

3. In the paper titled “GAP: Differentially Private Graph Neural Networks with Aggregation Perturbation” which was referenced in this paper, the authors have conducted a “Resilience Against Privacy Attacks” study in their experimental results to emphasize the resilience of their model against node membership inference attacks. Is it possible to perform similar experiments in this work?

**Limitations:**

Limitations are addressed.

---

> ### Author Rebuttal · Authors · 2024-08-05
>
> We are deeply grateful to Reviewer ApdU for the comprehensive feedback. Here, we will address these questions.
>
> >Edge DP is weak; consider node DP or stronger privacy notions. (W1)
>
> We sincerely appreciate your suggestion to explore node-level differential privacy as a potential extension of our work, a direction we agree could indeed enrich the scope of our research, as noted in the "Societal Impact and Limitations" section. However, we respectfully emphasize that edge DP still holds significant value within the private graph learning community, as recognized by both Reviewer y6QA and Reviewer sGeR. Furthermore, the previous work [1] most relevant to this work also focuses exclusively on edge DP for personalized Pagerank, underscoring its significance and practical applications. Our study specifically addresses edge DP within the context of graph diffusion.
>
> [1] Epasto et al. Differentially private graph learning via sensitivity-bounded personalized pagerank.
>
> >Iterative privacy framework and PageRank are well-known; lacks graph-specific insights. (W2)
>
> We respectfully disagree with the assessment. Two things being well-known does not mean building a connection between them is well known, let alone the algorithm having to be designed specifically by taking into account the features from both sides.
> First and foremost, our main contribution is the first one to introduce Privacy Amplification by Iteration analysis [1] within the graph analysis and graph learning community, to the best of our knowledge, while prior research has predominantly concentrated on optimization [1,2] and sampling [3]. Regarding the work by Thakurta et al., we did not find a work first-authored by Dr. Thakurta and closely related to the iterative privacy framework. We guess Reviewer ApdU might be referring to [1]. However, the applications of PABI to graph diffusion scenarios are not standard and require many specific designs to incorporate graph properties:
>
> (1) The degree-dependent thresholding function design in our work is motivated to achieve better control over edge perturbation. This design definitely involves graph-specific design.
>
> (2) Since graph diffusions propagate in $\ell_1$ space, we are the first to explore the use of Laplace noise instead of Gaussian noise,  while Laplace noise has seldom been considered in previous PABI analyses. The superiority of our approach is demonstrated in Appendix Figure 9 (Left). This further involves graph-specific consideration.
>
> (3) Most importantly, we propose a novel $\infty$-Wasserstein distance tracking method, which improves upon previous PABI analyses and significantly tightens our privacy bound (as demonstrated in Figures 3 and 6), thereby making our algorithm actually practical. Additionally, this $\infty$-Wasserstein distance tracking method is generalizable to other metric spaces and may be of interest for addressing other related problems.
>
> [1] Feldman et al. Privacy amplification by iteration.
>
> [2] Altschuler et al. On the Privacy of Noisy Stochastic Gradient Descent for Convex Optimization.
>
> [3] Altschuler et al. Faster high-accuracy log-concave sampling via algorithmic warm starts.
>
> >Can the algorithm be extended to GNN learning/inference? (W3)
>
> Yes, our algorithm may be extended to some graph neural networks, such as the private version of SGC [1], where the gradient information does not need to be back propagated through the graph structure. During the training phase, the algorithm can be used to propagate features and generate private node embeddings, which can then be utilized to train a classifier. In the inference phase, applicable to both transductive and inductive settings, these features can be propagated, and the trained classifier can be employed for inference. However, for standard message passing NNs where the nonlinearity requires to backpropagate gradients across graph structures, we cannot use our framework because the backpropagation may leak private information. Note that some recent DP-GNN works such as [2] indeed adopt the framework without gradient  backpropagating over graphs.
>
> [1] Wu et al. Simplifying graph convolutional networks.
>
> [2] Sajadmanesh et al. GAP: Differentially Private Graph Neural Networks with Aggregation Perturbation.
>
> >Utility-privacy trade-offs for different “m” values (Q1)
>
> We thank the reviewer for this insightful question. When adjusting “$m$” from $0$ to larger values, our algorithm transitions from noisy to noiseless graph diffusion with output perturbation. Our experiments show that noisy iterations generally outperform output perturbation in utility. Slightly delaying noise injection “$m$” leads to minor performance variations (gains or losses) depending on the dataset and privacy requirements. While practitioners can explore different “$m$” values to optimize privacy-utility trade-offs, our results suggest that $m=0$ is already an effective choice for noise injection.
>
> >Can DP guarantees be extended to node embeddings in graph prediction tasks? (Q2)
>
> Yes, it is possible to directly extend our analysis to node embeddings with DP guarantees as long as the non-linear graph convolution operators satisfy certain constraints, such as Lipschitzness, and do not involve learnable parameters that require backpropagation through the graph diffusion.
>
> >Can similar privacy tests, like in the GAP paper, be conducted? (Q3)
>
> In GAP paper, node-level MIA were employed to assess privacy leakage from learnable weights in GNNs under node DP, in scenarios where strong adversaries exploit overfitting to classify members based on GNN confidence. However, our algorithm does not involve a learning model (no learnable weights or data features) while only generating/output graph diffusion vectors, making it improper to be directly tested in GAP paper’s settings. We greatly value the Reviewer’s suggestion on privacy auditing for our algorithm and agree that exploring effective MIA is a promising direction for future research.

---

### Author Rebuttal · Authors · 2024-08-05

We sincerely appreciate the valuable feedback and insightful comments from all our reviewers. All reviewers recognize the significance of the problem we are addressing. Our work has contributed novel theoretical insights coupled with comprehensive empirical evaluations, which were particularly appreciated by Reviewer y6QA, Reviewer dwyJ, and Reviewer sGeR. Some questions remain regarding the generalization of our framework to other types of diffusions and its potential extension to graph neural networks, as well as the sensitivity of the thresholding parameter $\eta$, and the scalability of our algorithm. We are eager to provide further clarifications on these points in the detailed responses to each reviewer below.

---

### Decision · Program_Chairs · 2024-09-25

**Decision:**

Accept (poster)

**Comment:**

While the reviews of the paper varied a lot in their scores, I think the contribution towards the improvement of amplification by iteration in terms of reducing the dependence on diameter is significant. However, I will request the authors to carefully address the concerns regarding the notion of privacy (e.g., edge differential privacy), and other associated concerns regarding the motivation of the problem, in a future version of the paper.